# Jigsaw Game: Federated Clustering

**Jinxuan Xu**                                        *jinxuan.xu@rutgers.edu*
*Department of Electrical and Computer Engineering*
*Rutgers University*

**Hong-You Chen**                                        *chen.9301@osu.edu*
*Department of Computer Science and Engineering*
*The Ohio State University*

**Wei-Lun Chao**                                        *chao.209@osu.edu*
*Department of Computer Science and Engineering*
*The Ohio State University*

**Yuqian Zhang**                                        *yqz.zhang@rutgers.edu*
*Department of Electrical and Computer Engineering*
*Rutgers University*

**Reviewed on OpenReview:** *https://openreview.net/forum?id=8YcUJbxmmC*

## Abstract

Federated learning has recently garnered significant attention, especially within the domain of supervised learning. However, despite the abundance of unlabeled data on end-users, unsupervised learning problems such as clustering in the federated setting remain under-explored. In this paper, we investigate the federated clustering problem, with a focus on federated $k$-means. We outline the challenge posed by its non-convex objective and data heterogeneity in the federated framework. To tackle these challenges, we adopt a new perspective by studying the structures of local solutions in $k$-means and propose a one-shot algorithm called FeCA (Federated Centroid Aggregation). FeCA adaptively refines local solutions on clients, then aggregates these refined solutions to recover the global solution of the entire dataset in a single round. We empirically demonstrate the robustness of FeCA under various federated scenarios on both synthetic and real-world data. Additionally, we extend FeCA to representation learning and present DeepFeCA, which combines Deep-Cluster and FeCA for unsupervised feature learning in the federated setting.

## 1 Introduction

Federated learning (FL) has emerged as a promising framework, enabling model training across decentralized data. This approach addresses data privacy concerns by allowing data to remain on individual clients. The goal of FL is to collaboratively train a model across multiple clients without directly sharing data. Within this context, FedAvg McMahan et al. (2017) has been considered the standard approach in FL, designed to obtain a centralized model by averaging the models trained independently on each client's data.

Although FL has seen widespread applications in the domain of supervised learning, particularly in tasks like classification (Oh et al., 2021; Jiménez-Sánchez et al., 2023), its utilization in the unsupervised learning sphere is still largely unexplored, even though it holds significant potential and applicability in numerous practical situations. A notable example is the large collections of unlabeled photographs owned by most smartphone users. In such instances, federated unsupervised learning can be a powerful paradigm, enabling the use of unsupervised learning approaches to leverage the "collective wisdom" of these unlabeled data while safeguarding user privacy.

In this paper, we investigate federated unsupervised learning, particularly focusing on the popular clustering problem of $k$-means. In prior studies, clustering methods have been applied in FL mainly focusing on problems such as client selection (Ghosh et al., 2020; Long et al., 2023) and privacy enhancement (Li et al., 2022; Elhussein & Gürsoy, 2023), without a deep investigation into the unsupervised learning aspect. Moreover, existing distributed clustering methods overlook the unique challenges in FL, such as data heterogeneity and communication efficiency, making it difficult to apply in the federated setting. Our study extends to federated clustering, incorporating unsupervised clustering on individual clients within a federated framework.

One key challenge of federated clustering is the inherent non-convexity of clustering problems, presenting multiple equivalent global solutions and potentially even more local solutions. Standard algorithms like Lloyd's algorithm Lloyd (1982) can only find a local solution of the $k$-means problem, without guaranteeing global optimum. *We note that the term "local solution" in this context refers to a local optimal in optimization, not the solution learned from a client*[1]. This challenge is amplified in the federated setting, where each client's data is a distinct subset of the entire dataset. Even under the IID data sample scenario, each client's clustering results might be suboptimal local solutions containing spurious centroids far from the true global centroids. And this issue could become even more pronounced under non-IID scenarios.

To this end, we propose a one-shot federated $k$-means algorithm: Federated Centroid Aggregation (FeCA), offering a new approach by exploiting structured local solutions. In the $k$-means problem, local solutions carry valuable information from the global solution. The proposed algorithm resolves these local solutions and leverages their benign properties within the federated clustering framework. FeCA is built upon theoretical studies (Qian et al., 2021; Chen et al., 2024) derived in a centralized setting, which suggests that every local solution is structured and contains nontrivial information about the global solution. Specifically, a local solution consists of estimates of the $k$ ground truth centers, with a subset of these estimates being accurate.

One common concern of FL lies in the potential decrease in performance compared to centralized models due to data heterogeneity across clients. However, from the perspective of local solutions, federated clustering could benefit from the decentralized framework. Each client's solution, whether a local optimum or not, carries partial information about the global solution of the entire dataset. By incorporating multiple clients' solutions, the central server could potentially recover the global optimal solution in one shot, akin to assembling a *jigsaw* puzzle of clients' solutions. For instance, if a true centroid is missing from one client's solution, it might be identified in the solutions of other clients.

Therefore, FeCA is designed to recover the global solution for $k$-means clustering in a federated setting by refining and aggregating solutions from clients. First, Lloyd's algorithm for $k$-means is performed on each client's data. Then, FeCA adaptively refines spurious centroids using their structural properties to obtain a set of refined centroids for each client. Then refined centroids are sent to the central server, where FeCA aggregates them to recover the global solution of the entire dataset. By exploiting the structure in local solutions, FeCA is able to accurately identify the true $k$ centroids of the entire dataset in one shot.

We further extend FeCA beyond a pre-defined feature space to the modern deep feature framework (Liu et al., 2021). Specifically, we present DeepFeCA, a federated representation learning algorithm from decentralized unlabeled data. Concretely, we pair FeCA with clustering-based deep representation learning models such as DeepCluster Caron et al. (2018; 2020), which assign pseudo-labels according to $k$-means clustering and then train the neural network in a supervised manner. The resulting algorithm, DeepFeCA, alternates between applying FeCA to the current features and using DeepCluster for further training. This iterative process enhances the model's ability to learn meaningful representations from the decentralized data.

We evaluate both FeCA and DeepFeCA on benchmark datasets, including S-sets Fränti & Sieranoja (2018), CIFAR Krizhevsky et al. (2009), and Tiny-ImageNet Le & Yang (2015). FeCA consistently outperforms baselines in various federated settings, demonstrating its effectiveness in recovering the global solution. Furthermore, DeepFeCA shows promising performance in federated representation learning.

---

[1]For clarity, throughout this paper, we use the client's solution for the result obtained from a client. If the solution happens to be a local solution, we name it the client's local solution.

## 2 Related Work

**Federated learning.** Mainstream FL algorithms (McMahan et al., 2017; Khaled et al., 2020; Haddadpour & Mahdavi, 2019) adopt coordinate-wise averaging of the weights from clients. However, given the limited performance of direct averaging, other approaches have been proposed: Yurochkin et al. (2019); Wang et al. (2020); Zhang et al. (2023b); Tan et al. (2023) identify the permutation symmetry in neural networks and then aggregate after the adaptive permutation; Lin et al. (2020); He et al. (2020); Zhou et al. (2020); Chen & Chao (2021); Zeng et al. (2023) replace weight average by model ensemble and distillation. These studies enhance the performance of the synchronization scheme but overlook the impact of local solutions on clients.

**Federated Clustering.** Many distributed clustering methods (Balcan et al., 2013; Bachem et al., 2018; Kargupta et al., 2001; Januzaj et al., 2004; Hess et al., 2022) have been proposed, but they overlook the heterogeneous challenge in FL. For synchronizing results returned from different clustering solutions, consensus clustering has been studied widely (Monti et al., 2003; Goder & Filkov, 2008; Li et al., 2021). But it works on the same dataset, unlike FL. In the context of FL, Qiao et al. (2021); Li et al. (2023); Xia et al. (2020); Lu et al. (2023); Li et al. (2022) focus on communication efficiency or privacy-preserving. A recent federated clustering study Stallmann & Wilbik (2022) proposes weighted averaging for Fuzzy $c$-means but requires multiple rounds. The study most relevant to ours introduces $k$-FED Dennis et al. (2021), a one-shot federated clustering algorithm, under a rather strong assumption that each client only has data from a few true clusters. It is still underexplored for federated clustering and usage of local solutions.

**Federated representation learning.** FEDREP Collins et al. (2021) studies supervised representation learning by alternating updates between classifiers and feature extractors. Jeong et al. (2020); Zhang et al. (2020) study federated semi-supervised learning with the server holding some labeled data and clients having unlabeled data. For federated unsupervised learning, Zhuang et al. (2021) proposes self-supervised learning in non-IID settings with a divergence-aware update strategy for mitigating non-IID challenges, distinct from our clustering focus. Zhang et al. (2023a) adopts the contrastive approach for model training on clients. A recent framework Lubana et al. (2022) introduces federated unsupervised learning with constrained clustering for representation learning, while our focus lies on exploring federated clustering via local solutions.

## 3 Background

**Clustering.** Given a $d$-dimensional dataset $\mathcal{D} = \{x_1 \in \mathbb{R}^d, \ldots, x_N \in \mathbb{R}^d\}$, the goal of $k$-means problem is to identify $k$ centroids $\mathcal{C} = \{c_1 \in \mathbb{R}^d, \ldots, c_k \in \mathbb{R}^d\}$ that minimize the following objective

$$G(\mathcal{C}) \doteq \sum_{n=1}^{N} \min_{j \in [k]} \|x_n - c_j\|_2^2. \tag{1}$$

**Federated clustering.** In the federated setting, the dataset $\mathcal{D}$ is decentralized across $M$ clients. Each client $m \in [M]$ possesses a distinct subset $\mathcal{D}_m$ of the entire dataset $\mathcal{D}$. Despite different data configurations, the goal of federated clustering remains the same – to identify $k$ centroids $\mathcal{C} = \{c_1, \ldots, c_k\}$ for $\mathcal{D} = \cup_m \mathcal{D}_m$. Under this federated framework, the optimization problem in Equation 1 can be reformulated as

$$\min_{\mathcal{C}} \ G(\mathcal{C}) = \sum_{m=1}^{M} G_m(\mathcal{C}), \tag{2}$$

where $G_m$ is the $k$-means objective computed on $\mathcal{D}_m$. Due to privacy concerns that restrict direct data sharing among clients, the optimization problem described in Equation 2 cannot be solved directly. Thus, the proposed algorithm FECA utilizes a collaborative approach between clients and a central server. Initially, each client $m$ independently minimizes $G_m(\mathcal{C})$ to obtain a set of $k$ centroids $\mathcal{C}^{(m)} = \{c_1^{(m)}, \ldots, c_k^{(m)}\}$ from their dataset $\mathcal{D}_m$. Then the server aggregates centroids $\cup_m \mathcal{C}^{(m)}$ to find a set of $k$ centroids $\mathcal{C}$ for $\mathcal{D}$.

We note that when clients perform standard Lloyd's algorithm for $k$-means clustering, they usually end up with local solutions $\mathcal{C}^{(m)}$, resulting in suboptimal performance even with IID distributed data $\mathcal{D}_m$. These local solutions can significantly complicate the aggregation process on the central server. Thus, the key

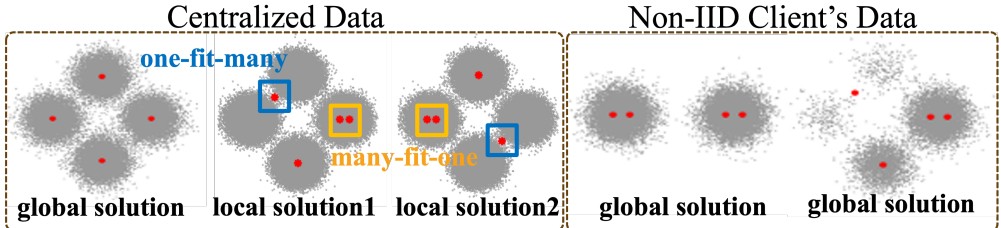

Figure 1: **Clustering results.** (Left): global and local solutions on centralized/IID client's data; (Right): global solutions for non-IID client's data sharing similar structures.

challenge in federated clustering lies in effectively resolving the client's local solutions and appropriately aggregating them on the central server.

### 3.1 Structure of Local Solutions

To better resolve the federated clustering problem, we propose to take a deeper look at local solutions in $k$-means, which often significantly differ from the global minimizer. Recent theoretical works by Qian et al. (2021); Chen et al. (2024) have established a positive result that under certain separation conditions, all the local solutions share a common geometric structure. More formally, suppose a local solution identifies centroids $\{c_1, \ldots, c_k\}$. Then, there exists a one-to-one association between these centroids and the true centers $\{c_1^*, \ldots, c_k^*\}$ from the global solution. This association ensures that each centroid $c_i$ belongs to exactly one of the following cases with overwhelming probability[2]:

- **Case 1** (one-fit-many association): centroid $c_i$ is associated with $s$ ($s > 1$) true centers $\{c_{j_1}^*, \ldots, c_{j_s}^*\}$.
- **Case 2** (one/many-fit-one association): $t$ ($t \geq 1$) centroids $\{c_{i_1}, \ldots, c_{i_t}\}$ are all associated with one true center $c_j^*$.

Namely, a centroid $c_i$ in a local solution is either a *one-fit-many* centroid that is located in the middle of multiple true centers (case 1, when $s > 1$), or a *one/many-fit-one* centroid that is close to a true center (case 2). Notably, when $c_i$ is the only centroid near a true center (case 2, when $t = 1$), it is considered a correctly identified centroid that closely approximates a true center. An illustration is provided in Figure 1. Next, we will introduce how our algorithm utilizes such local solution structures to obtain unified clustering results in the federated setting.

## 4  Jigsaw Game – FeCA

---
**Algorithm 1** Federated Centroid Aggregation (FECA)
---
1: **input** cluster number $k$
2: **for** each client $m = 1, \ldots, M$ **do**
3:     $\mathcal{C}^{(m)}, \mathcal{D}^{(m)} \leftarrow$ ClientUpdate $(k)$
4:     $\mathcal{R}^{(m)} \leftarrow$ RadiusAssign $(\mathcal{C}^{(m)}, \mathcal{D}^{(m)})$
5: **end for**
6: $\mathcal{C} \leftarrow \bigcup_{m=1}^{M} \mathcal{C}^{(m)}$
7: $\mathcal{R} \leftarrow \bigcup_{m=1}^{M} \mathcal{R}^{(m)}$
8: $\mathcal{C}^* \leftarrow$ ServerAggregation $(k, \mathcal{C}, \mathcal{R})$
9: **return** $\mathcal{C}^*$
---

The proposed federated clustering algorithm FECA is built upon the collaboration between clients and a central server. Each client $m$ shares its refined centroid solution $\mathcal{C}^{(m)}$ with the server, where each $\mathcal{C}^{(m)}$ carries partial information of the global solution, similar to pieces of a jigsaw puzzle. The server then aggregates

---
[2]Such structure of local solutions holds even when $k \neq k^*$, where $k^*$ is the number of true clusters in the dataset.

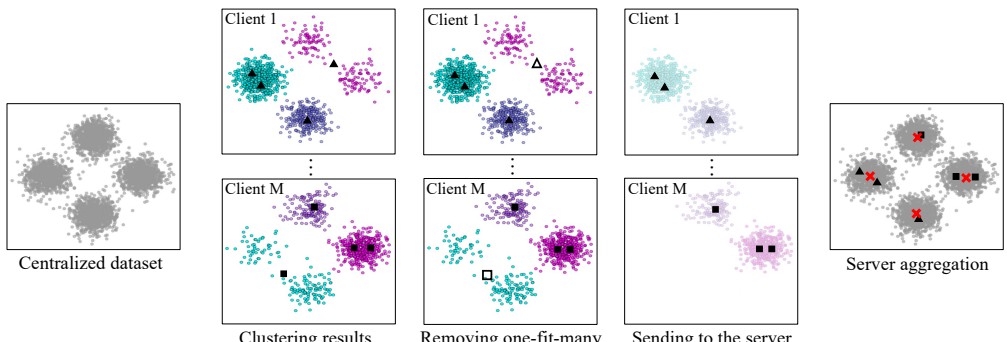

Figure 2: **FeCA roadmap.** 1st column: The centralized dataset distributed to clients. 2nd column: The $k$-means clustering results on different clients under non-IID data sample scenario, where black triangles and squares represent centroids. 3rd column: Eliminating one-fit-many centroids in Algorithm 2, indicated by hollow squares and triangles. 4th column: Centroids sent to the server. 5th column: Aggregation of received centroids on the server where red crosses represent recovered centroids.

received centroids $\cup_m \mathcal{C}^{(m)}$ to obtain a unified complete solution $\mathcal{C}^*$, akin to assembling puzzle pieces in a *jigsaw game*. FeCA only requires one communication between clients and the server, thanks to its adaptive refinement of local solutions on the client side. The detailed procedure is presented in Algorithm 1 and illustrated in Figure 2.

**Privacy concern.** While privacy is crucial in FL, it is not our main focus. However, an advantage of our one-shot algorithm is its minimal information exchange compared to standard iterative approaches like distributed clustering. In FeCA, sending refined centroids to the server is viewed as no more privacy risk than mainstream FL approaches of sending models with classifiers, which more or less convey class or cluster information.

## 4.1 Client Update Algorithm

---

**Algorithm 2** FeCA- ClientUpdate

---

1: **input** cluster number $k$
2: Apply Lloyd's algorithm on the data of client $m$ to obtain $k$ clusters $\mathcal{D}^{(m)} = \{\mathcal{D}_1^{(m)}, \ldots, \mathcal{D}_k^{(m)}\}$ and corresponding $k$ centroids $\mathcal{C}^{(m)} = \{c_1^{(m)}, \ldots, c_k^{(m)}\}$.
3: **while** $\mathcal{C}^{(m)} \neq \oslash$ **do**
4:      Detect one-fit-many centroid $c_i^{(m)}$ whose cluster $\mathcal{D}_i^{(m)}$ has the largest standard deviation and calculate the objective value $G_i^{(m)}$ of the cluster $\mathcal{D}_i^{(m)}$.
5:      Detect many-fit-one centroids $c_p^{(m)}$, $c_q^{(m)}$ of clusters $\mathcal{D}_p^{(m)}$ and $\mathcal{D}_q^{(m)}$ with the smallest pairwise distance.
6:      Merge $\mathcal{D}_p^{(m)}$ and $\mathcal{D}_q^{(m)}$ into one cluster $\mathcal{D}_j^{(m)}$ to obtain a new centroid $c_j^{(m)}$ and calculate the objective value $G_j^{(m)}$ of the merged cluster $\mathcal{D}_j^{(m)}$.
7:      **if** $G_i^{(m)} \geq G_j^{(m)}$ **then**
8:          $\mathcal{C}^{(m)} \leftarrow \mathcal{C}^{(m)} \setminus \{c_i^{(m)}\}$
9:          $\mathcal{D}^{(m)} \leftarrow \mathcal{D}^{(m)} \setminus \{\mathcal{D}_i^{(m)}\}$
10:      **else**
11:          Terminate the loop.
12:      **end if**
13: **end while**
14: **return** centroids $\mathcal{C}^{(m)}$, clusters $\mathcal{D}^{(m)}$

---

This step aims to refine the spurious local solutions of $k$-means clustering on clients. Each client $m$ first performs standard Lloyd's algorithm to obtain a set of $k$ centroids $\mathcal{C}^{(m)} = \{c_1^{(m)}, \ldots, c_k^{(m)}\}$, and this solution is only guaranteed to be a local solution. As discussed in subsection 3.1, despite the variations in solutions across different clients, each $\mathcal{C}^{(m)}$ always possesses some centroids (one/many-fit-one) that are proximate to a subset of ground truth centers. To facilitate the aggregation process on the server, we propose retaining only centroids from $\mathcal{C}^{(m)}$ that are positioned close to true centers.

The client update step in FECA focuses on refining the solution $\mathcal{C}^{(m)}$ by eliminating one-fit-many centroids that are distant from any true center. Specifically, a one-fit-many centroid is always located in the middle of multiple nearby true clusters, making it distant from most data points in those clusters and leading to a high standard deviation for its cluster. Conversely, many-fit-one centroids, which fit the same true center, are close to each other and thus have a small pairwise distance. As presented in Algorithm 2, we first use these properties to detect the candidate one-fit-many $c_i^{(m)}$ and many-fit-one $c_p^{(m)}, c_q^{(m)}$ centroids, which are likely from a spurious local solution.

Next, for further refinement, we need to confirm if these candidate centroids indeed originate from a local solution. To this end, for detected one-fit-many centroid $c_i^{(m)}$, we first calculate the objective value $G_i^{(m)}$ of its cluster $\mathcal{D}_i^{(m)}$ as

$$G_i^{(m)} = \sum\nolimits_{x \in \mathcal{D}_i^{(m)}} \|x - c_i^{(m)}\|_2^2. \tag{3}$$

For detected many-fit-one centroids, we merge their clusters $\mathcal{D}_p^{(m)}, \mathcal{D}_q^{(m)}$ to form a new cluster $\mathcal{D}_j^{(m)}$ with the corresponding centroid $c_j^{(m)}$. And then we calculate the objective value $G_j^{(m)}$ as

$$G_j^{(m)} = \sum\nolimits_{x \in \{\mathcal{D}_p^{(m)} \cup \mathcal{D}_q^{(m)}\}} \|x - c_j^{(m)}\|_2^2. \tag{4}$$

If the current solution $\mathcal{C}^{(m)}$ is only locally optimal, $\mathcal{D}_i^{(m)}$ should contain data from multiple true clusters with a large $G_i^{(m)}$, while $\mathcal{D}_j^{(m)}$ only contains data from one true cluster with a small $G_j^{(m)}$. Therefore, if $G_i^{(m)}$ is greater than $G_j^{(m)}$, it confirms that these candidate centroids stem from a local solution. In such cases, Algorithm 2 removes $c_i^{(m)}$ from $\mathcal{C}^{(m)}$ for not being close to any true center. Otherwise, if $G_i^{(m)}$ is less than $G_j^{(m)}$, these centroids are regarded as the correct portion with no need for further refinement.

Notably, there may be multiple groups of centroids that possess the local structure. The algorithm is designed to iteratively identify and refine the local solution. Our theoretical analysis (Lemma A.1 in the appendix) demonstrates that Algorithm 2 effectively removes all one-fit-many centroids from local solutions under the Stochastic Ball Model. In this model, we assume that each client's data is sampled independently and uniformly from one of $k$ disjoint balls centered at the ground truth centers. A formal definition is provided in Appendix A.

## 4.2 Radius Assign Algorithm

After removing one-fit-many centroids in the ClientUpdate phase, only centroids near true centers (one/many-fit-one) would be sent to the server. This RadiusAssign step prepares these centroids for server-side aggregation by assigning a specific radius to each. This setup allows the server to utilize these radii for effective aggregation. The primary goal of this step is to determine the radius that best approximates the true cluster radius of the entire dataset. In this section, we present two algorithmic variants for the RadiusAssign step. The first variant Algorithm 3 is designed for *theoretical* validation purposes, while the second variant Algorithm 4 is tailored for *empirical* experimentation.

Through the theoretical variant, we establish Theorem 4.1 that characterizes the performance of our algorithm FECA under the Stochastic Ball Model. This theoretical variant generates a tentative solution $\tilde{\mathcal{C}}^{(m)}$ by discarding any potential many-fit-one centroids within $\mathcal{C}^{(m)}$. Following this, a radius $r^{(m)}$ is then calculated according to the minimum pairwise distance among centroids in $\tilde{\mathcal{C}}^{(m)}$, and this radius is assigned to every centroid in the original solution $\mathcal{C}^{(m)}$ from Algorithm 2. We note that the method for identifying many-fit-one centroids utilized in this theoretical variant is only applicable under the Stochastic Ball Model.

---

**Algorithm 3** FECA- RadiusAssign (*Theoretical*)

---

1: **input** centroids $\mathcal{C}^{(m)}$, clusters $\mathcal{D}^{(m)}$
2: Initialize a new set of centroids $\tilde{\mathcal{C}}^{(m)} = \mathcal{C}^{(m)}$.
3: Determine $r'_i$ for each centroid $\tilde{c}_i^{(m)} \in \tilde{\mathcal{C}}^{(m)}$ as the maximum distance from any data point in $\mathcal{D}_i^{(m)}$ to $\tilde{c}_i^{(m)}$: $r'_i = \max_{x_i^{(m)} \in \mathcal{D}_i^{(m)}} \|x_i^{(m)} - \tilde{c}_i^{(m)}\|_2$.
4: **for** each centroid $\tilde{c}_i^{(m)} \in \tilde{\mathcal{C}}^{(m)}$ **do**
5:     Identify the nearest centroid $\tilde{c}_{j\neq i}^{(m)} \in \tilde{\mathcal{C}}^{(m)}$ to $\tilde{c}_i^{(m)}$ with the pairwise distance denoted as $d_{ij}$.
6:     **if** $r'_i + r'_j > d_{ij}$ **then**
7:         $\tilde{\mathcal{C}}^{(m)} \leftarrow \tilde{\mathcal{C}}^{(m)} \setminus \{\tilde{c}_i^{(m)}, \tilde{c}_j^{(m)}\}$
8:     **end if**
9: **end for**
10: Calculate the minimal pairwise distance between centroids in $\tilde{\mathcal{C}}^{(m)}$: $\tilde{\Delta}_{min}^{(m)} = \min_{\tilde{c}_i, \tilde{c}_j \in \tilde{\mathcal{C}}^{(m)}, i \neq j} \|\tilde{c}_i - \tilde{c}_j\|_2$.

11: Determine the uniform radius $r^{(m)} = \frac{1}{2}\tilde{\Delta}_{min}^{(m)}$ and assign it to each centroid $c_i^{(m)} \in \mathcal{C}^{(m)}$.
12: $\mathcal{R}^{(m)} \leftarrow \bigcup_i \left(c_i^{(m)}, r^{(m)}\right)$
13: **return** $\mathcal{R}^{(m)}$

---

**Algorithm 4** FECA- RadiusAssign (*Empirical*)

---

1: **input** centroids $\mathcal{C}^{(m)}$, clusters $\mathcal{D}^{(m)}$
2: Determine $r'_i$ for each centroid $c_i^{(m)} \in \mathcal{C}^{(m)}$ as the maximum distance from any data point in $\mathcal{D}_i^{(m)}$ to $c_i^{(m)}$: $r'_i = \max_{x_i^{(m)} \in \mathcal{D}_i^{(m)}} \|x_i^{(m)} - c_i^{(m)}\|_2$.
3: **for** each centroid $c_i^{(m)} \in \mathcal{C}^{(m)}$ **do**
4:     Identify the nearest centroid $c_{j\neq i}^{(m)} \in \mathcal{C}^{(m)}$ to $c_i^{(m)}$ with the pairwise distance denoted as $d_{ij}$, and then calculate $r''_i = \frac{1}{2}d_{ij}$.
5:     Determine the unique radius $r_i^{(m)} = \min(r'_i, r''_i)$ and assign it to the centroid $c_i^{(m)}$.
6: **end for**
7: $\mathcal{R}^{(m)} \leftarrow \bigcup_i \left(c_i^{(m)}, r_i^{(m)}\right)$
8: **return** $\mathcal{R}^{(m)}$

---

However, in real-world applications, especially under non-IID data sample scenarios, it is both challenging and unnecessary to eliminate all many-fit-one centroids from clients' solutions, as they often align closely with true centers. Accordingly, we develop an empirical variant, Algorithm 4, which assumes only one-fit-many centroids are excluded and assigns a unique radius $r_i^{(m)}$ to each centroid $c_i^{(m)} \in \mathcal{C}^{(m)}$. As for remaining many-fit-one centroids, their radii are estimated as half of their pairwise distances, which are typically much smaller compared to those of correct centroids. The server then groups all received centroids based on these radii, prioritizing the largest ones first. This ensures that the smaller radii associated with many-fit-one centroids minimally impact the aggregation process. An in-depth analysis of Algorithm 4 is provided in Appendix B, showcasing its effectiveness across a variety of experimental settings, including those with high data heterogeneity.

It is worth noting that the theoretical variant is designed for theoretical analysis under the Stochastic Ball Model assumption. This assumption enables easy identification of many-fit-one centroids for clearer cluster separation approximation and accurate radius assignment. In contrast, the empirical variant does not need to remove many-fit-one centroids, as they are close to true centers and aid in reconstructing the global solution on the server side. This approach allows the empirical variant to assign distinct radii to each remaining centroid, enhancing the algorithm's effectiveness and practicality without relying on limited assumptions. A detailed comparison of the theoretical and empirical variants is provided in subsection C.4.

### 4.3 Server Aggregation Algorithm

---

**Algorithm 5** FECA- ServerAggregation

---

1: **input** cluster number $k$, centroids $\mathcal{C}$, radius $\mathcal{R}$
2: $n = 1$
3: **while** $\mathcal{C} \neq \oslash$ **do**
4:     Pick $(c_i, r_i) \in \mathcal{R}$ with the largest $r_i$.
5:     Let $\mathcal{S}_n = \{c_t : c_t \in \mathcal{C}, \|c_t - c_i\|_2 \leq r_i\}$
6:     Set $\mathcal{C} \leftarrow \mathcal{C} \setminus \mathcal{S}_n$
7:     $n = n + 1$
8: **end while**
9: Select top $k$ sets $\mathcal{S}_n$ containing the largest number of elements.
10: $\mathcal{C}^* \leftarrow \text{mean}(\mathcal{S}_j), j \in [k]$
11: **return** $\mathcal{C}^*$

---

At the server stage, the goal is to aggregate all received centroids $\mathcal{C} = \{\mathcal{C}^{(1)}, \ldots, \mathcal{C}^{(M)}\}$ from $M$ clients into a unified set of $k$ centroids $\mathcal{C}^*$. This task presents apparent challenges: due to the preceding refinement stage, clients may contribute varying numbers of centroids, and the indices of these centroids often lack consistency across clients. However, assuming the refinement phase in Algorithm 2 effectively removes spurious one-fit-many centroids far from true centers, the returned centroids on the server would be closely grouped around true centers. This phenomenon enables a straightforward classification of all returned centroids into $k$ distinct groups, each aligned with one of the $k$ true centers, as presented in Algorithm 5. Equivalently, this is another clustering problem based on returned centroids under a high Signal-to-Noise Ratio (SNR) separation condition. Finally, the server calculates the means of centroids within each group to obtain $\mathcal{C}^*$.

In some extreme cases where the number of groups $n$ might be less than $k$, such as when all clients converge to the same local solution. In such cases, Algorithm 2 removes one-fit-many centroids associated with the same true clusters from all clients. This renders it impossible for Algorithm 5 to reconstruct corresponding true centers without receiving any associated centroids from clients. It is important to note that this scenario is trivial within the federated framework, where all clients share the same local solutions. Essentially, it is akin to having only one client encountering a local solution. Further discussion on cases when $n < k$ is provided in the Appendix E.

### 4.4 Theoretical Analysis

We now state our main theorem, which characterizes the performance of FECA under the Stochastic Ball Model. Assume the data $x^{(m)}$ of client $m$ is sampled independently and uniformly from one of $k$ disjoint balls $\mathbb{B}_s$ with radius $r$, each centered at a true center $\theta_s^*$, $s \in [k]$. Each ball component under the Stochastic Ball Model has a density

$$f_s(x) = \frac{1}{\text{Vol}(\mathbb{B}_s)} \mathbb{1}_{\mathbb{B}_s}(x). \tag{5}$$

Additionally, we define the maximum and minimum pairwise separations between the true centers $\{\theta_s^*\}_{s \in [k]}$ as

$$\Delta_{max} := \max_{s \neq s'} \|\theta_s^* - \theta_{s'}^*\|_2, \quad \Delta_{min} := \min_{s \neq s'} \|\theta_s^* - \theta_{s'}^*\|_2.$$

**Theorem 4.1.** *(Main Theorem) Under the Stochastic Ball Model, for some constants $\lambda \geq 3$ and $\eta \geq 5$, if*

$$\Delta_{max} \geq 4\lambda^2 k^4 r \quad and \quad \Delta_{min} \geq 10\eta\lambda k^2 \sqrt{r\Delta_{max}},$$

*then by utilizing the radius determined by Algorithm 3, any output centroid $c_s^*$ from Algorithm 1 is close to some ground truth center:*

$$\|c_s^* - \theta_{s'}^*\|_2 \leq \frac{4}{5\eta}\Delta_{min}. \tag{6}$$

Theorem 4.1 characterizes the performance of our main algorithm FECA, utilizing the radius from Algorithm 3. The proof, provided in Appendix A, builds on the infinite-sample and high SNR assumptions established in Qian et al. (2021) which characterizes local solutions of centralized $k$-means. Next, we will provide a discussion of both conditions.

- **Separation Condition:** the separation between true centers $\Delta_{min}$ and $\Delta_{max}$ cannot be too small is generally necessary for a local solution to bear the structural properties described in subsection 3.1 Qian et al. (2021). Additionally, the ratio $\frac{\Delta_{max}}{\Delta_{min}}$ indicates how evenly spaced the true centers are, with the ratio approaching 1 when the true centers are nearly evenly spaced.

- **Technical Assumptions:** our main theorem heavily depends on *the Stochastic Ball Model* and *infinity sample* assumptions. We would love to note that the local solution structure also holds when the data follows the Gaussian mixture model or has finite data samples Chen et al. (2024). We view these technical assumptions as less important than the above separation condition and will corroborate using both synthetic and real clustering data to demonstrate the effectiveness of our algorithm.

Note that above assumptions are often not met in practice. Thus, we develop another variant, Algorithm 4, which does not require the elimination of many-fit-one centroids and assigns a unique radius to each returned centroid from the client. A detailed empirical evaluation of these radii determined by Algorithm 4 is presented in Appendix B, showcasing their effectiveness in supporting our algorithm FECA.

### 4.5 Discussions on Heterogeneity

We assume that the client's local solution for its dataset $\mathcal{D}_m$ is also a local solution of the entire dataset $\mathcal{D} = \cup_m \mathcal{D}_m$, which allows us to leverage the structures discussed in subsection 3.1. This assumption holds when $\mathcal{D}_m$ is an IID-sampled subset from $\mathcal{D}$. Our experiments showcase our algorithm's robustness even under non-IID conditions. Here we provide an explanation in Figure 1, where right plots illustrate two clients' non-IID sampled data and the corresponding *global* solutions (achieve a global optimum when $k = k^{*3}$). We found that despite the data heterogeneity, the clients' global solutions share similar structures as described in subsection 3.1. We attribute these observations to the fact that under non-IID conditions, clients' data tend to concentrate on some of the true clusters. This increases the chance that clients' global solutions contain many-fit-one centroids for those true clusters, which can be aggregated together on the server by our algorithm FECA. This scenario also suggests that even if a client can recover the global solution on its non-IID data, such a global solution coincides with a local solution on IID data and we still need to deploy FECA to produce the final solution.

## 5 DeepFeCA

With FECA, we can learn centroids $\mathcal{C}^*$ in the pre-defined feature space collaboratively with multiple clients. In this section, we extend FECA to unsupervised representation learning Liu et al. (2021), aiming to learn a feature extractor $f_{\boldsymbol{\theta}}$ parameterized by $\boldsymbol{\theta}$ from the unlabeled data set $\mathcal{D}$, such that the extracted feature $f_{\boldsymbol{\theta}}(x)$ of data $x$ can better characterize its similarity or dissimilarity to other data instances.

Some studies (Asano et al., 2020; Caron et al., 2020; 2018) integrate clustering approaches with unsupervised representation learning. For instance, Caron et al. (2018) proposed DEEPCLUSTER, which learns $f_{\boldsymbol{\theta}}(x)$ from an unlabeled dataset $\mathcal{D} = \{x_n\}_{n=1}^N$ by repeating two steps:

- Perform Lloyd's algorithm on $\{f_{\boldsymbol{\theta}}(x_n)\}_{n=1}^N$ to obtain a set of $k$ centroids $\mathcal{C} = \{c_j\}_{j=1}^k$;

- Create a pseudo-labeled set $\mathcal{D} = \{(x_n, \hat{y}_n)\}_{n=1}^N$ where $\hat{y}_n = \arg\min_j \|x_n - c_j\|_2$, and learn $f_{\boldsymbol{\theta}}$ with a linear classifier in a supervised fashion for multiple epochs.

---

[3] $k^*$ indicates the number of true centers in the entire dataset.

---

**Algorithm 6** DeepFeCA for Representation Learning

---

1: **Input:** cluster number $k$, total number of clients $M$, round number $T$
2: **Initialization:** server model $\bar{\boldsymbol{\theta}}$
3: **for** round $t = 1, \cdots, T$ **do**
4:     Perform FeCA with $M$ clients to obtain $k$ aggregated centroids $\mathcal{C}^*$ ($k$-means clustering is performed on features extracted by $f_{\bar{\boldsymbol{\theta}}}$ on each client).
5:     Broadcast $\bar{\boldsymbol{\theta}}$ and $\mathcal{C}^*$ to all clients.
6:     **for** each client $m \in [M]$ **do**
7:         Create $\mathcal{D}_m = \{(x_n^{(m)}, \hat{y}_n^{(m)})\}_{n=1}^{N_m}$ where $\hat{y}_n^{(m)} = \arg\min_j \|x_n^{(m)} - c_j\|_2$ and $c_j \in \mathcal{C}^*$.
8:         Train $\bar{\boldsymbol{\theta}}$ together with a linear classifier on $\mathcal{D}_m$ for multiple epochs to obtain the client model $\boldsymbol{\theta}^{(m)}$.
9:     **end for**
10:    $\bar{\boldsymbol{\theta}} \leftarrow \sum_{m=1}^{M} \frac{|\mathcal{D}_m|}{|\mathcal{D}|} \boldsymbol{\theta}^{(m)}$
11: **end for**
12: Return the model $\bar{\boldsymbol{\theta}}$

---

DeepCluster is known for its simplicity and has been shown to perform on par with other more advanced self-supervised learning methods for representation learning Ericsson et al. (2021). In this paper, we extend DeepCluster to the federated setting and propose DeepFeCA, which integrates DeepCluster within the FedAvg McMahan et al. (2017) framework. Namely, DeepFeCA iterates between local training of $f_{\boldsymbol{\theta}}(x)$ on each client and global model aggregation for multiple rounds. At the end of each round, DeepFeCA applies FeCA to update the centroids that are used to assign pseudo labels. Algorithm 6 outlines our approach, where the red text corresponds to FeCA, the blue text corresponds to DeepCluster, and the green text corresponds to the element-wise weight average of FedAvg.

## 6 Experiments

We first evaluate FeCA on benchmark synthetic datasets, which have well-established true centers. Then we extend our evaluation to frozen features of real-world image data extracted from pre-trained neural networks. Additionally, we assess the representation learning capabilities of DeepFeCA by training a deep feature extractor network from scratch in the federated framework.

To simulate the non-IID data partitions, we follow Hsu et al. (2019) to split the data drawn from Dirichlet($\alpha$) for multiple clients. Smaller $\alpha$ indicates that the split is more heterogeneous. We also include the IID setting, in which clients are provided with uniformly split subsets of the entire dataset. Furthermore, we have standardized the number of clients to $M = 10$ for all experiments in this section. A detailed discussion on the impact of varying the number of clients is provided in Appendix D.

**Baselines.** We mainly compare three baselines:

- **Match Averaging (M-Avg)**: matches different sets of centroids from clients by minimizing their $\ell_2$-distances and returns the means of matched centroids.

- $k$**-FED**: the one-shot federated clustering method Dennis et al. (2021) designed for heterogeneous data, assuming that each client's data originates from $k' \leq \sqrt{k^*}$ true clusters. The method utilizes a small $k'$ for $k$-means clustering on clients. In the following experiments, we select a single $k'$ for each dataset, with detailed tuning experiments provided in subsection C.5.

- **FFCM**: Stallmann & Wilbik (2022) focuses on fuzzy $c$-means clustering and presents two versions of aggregation algorithms, which are weighted averaging centroids (v1) and applying $k$-means on centroids (v2). Since this method is not designed for a one-shot setting, we report its results for both round 1 and round 10 in the following experiments.

Additionally, we include a **centralized** benchmark, representing the performance of $k$-means clustering on the entire dataset without federated splits. In the following experiments, we set $k = k^*$ for all methods except for $k$-FED.

**Evaluation metric.** For synthetic datasets with known true centers, we assess recovered centroids by calculating the $\ell_2$-distance between output centroids and true centers. In contrast, for real datasets where true centers are unknown, we adopt the standard clustering measures including *Purity* and *Normalized Mutual Information (NMI)*. The average Purity for all clusters is reported. NMI measures the mutual information shared between clustering assignments $X$ and true labels $Y$ defined as $NMI(X,Y) = 2I(X;Y)/(H(X) + H(Y))$, where $I$ denotes the mutual information and $H$ is the entropy.

## 6.1 FeCA Evaluation

**On synthetic datasets.** We evaluate FeCA on benchmark datasets in Fränti & Sieranoja (2018) with known true centers. Specifically, we focus on S-sets, comprising synthetic data characterized by four different degrees of separation. S-sets includes four sets: S1, S2, S3, and S4, each consisting of 15 Gaussian clusters in $\mathbb{R}^2$. Visualizations of S-sets are provided in Appendix C.

We assess recovered centroids by calculating the $\ell_2$-distance to ground truth centers, with mean results and standard deviation from 10 runs reported in Table 1. Additionally, the Purity and NMI of clustering assignment quality are presented in subsection C.1. We investigate three data sample scenarios in the federated setting: IID, Dirichlet(0.3), and Dirichlet(0.1). And we select $k' = 5$ for $k$-FED after careful tuning (detailed in subsection C.5).

Results in Table 1 indicate that our algorithm FeCA consistently outperforms all baselines in recovering the global solution across all experimental settings. Figure 4 provides a visualization of these results, demonstrating that even under the challenging non-IID scenario – Dirichlet(0.3), FeCA's recovered centroids closely approximate the true centers.

Table 1: $\ell_2$**-distance (scaled by $10^4$) between recovered centroids and true centers on S-sets under three data sample scenarios.** Dirichlet(0.3) and Dirichlet(0.1) are denoted as (0.3) and (0.1), respectively. *Rd* indicates rounds for FFCM. We report mean±std from 10 random runs.

| $\ell_2$-distance $\times 10^4$ | S-sets (S1) | | | S-sets (S2) | | |
|---|---|---|---|---|---|---|
| Method | IID | (0.3) | (0.1) | IID | (0.3) | (0.1) |
| M-Avg | 7.2±4.3 | 45.8±5.0 | 63.0±2.8 | 14.7±7.1 | 51.4±8.9 | 64.0±11.1 |
| FFCMv1(Rd=1) | 12.2±15.8 | 80.7±13.3 | 81.1±15.7 | 16.7±20.4 | 70.9±13.3 | 89.7±16.0 |
| FFCMv1(Rd=10) | 3.3±1.4 | 50.3±8.8 | 66.1±12.4 | 9.6±19.1 | 56.9±12.3 | 60.3±18.2 |
| FFCMv2(Rd=1) | 23.1±19.4 | 75.7±16.8 | 77.0±14.4 | 16.1±16.9 | 71.9±13.5 | 79.9±12.6 |
| FFCMv2(Rd=10) | 2.6±0.4 | 42.9±14.8 | 54.2±14.6 | 32.9±0.5 | 58.1±13.5 | 59.4±25.8 |
| $k$-FED($k'$=5) | 84.6±15.8 | 59.4±12.3 | 59.8±16.9 | 75.9±17.7 | 64.1±13.1 | 63.9±11.6 |
| FeCA | **1.0**±0.1 | **6.8**±3.7 | **22.3**±15.6 | **1.9**±0.1 | **13.6**±16.1 | **38.8**±3.4 |
| Centralized | | 14.3±18.7 | | | 8.4±14.7 | |
| $\ell_2$-distance $\times 10^4$ | S-sets (S3) | | | S-sets (S4) | | |
| Method | IID | (0.3) | (0.1) | IID | (0.3) | (0.1) |
| M-Avg | 16.4±6.8 | 35.5±5.7 | 46.7±5.3 | 14.2±3.7 | 28.7±3.3 | 40.1±6.4 |
| FFCMv1(Rd=1) | 16.6±15.2 | 58.1±11.9 | 77.4±8.9 | 11.4±9.5 | 45.0±8.9 | 63.1±10.8 |
| FFCMv1(Rd=10) | 11.4±15.5 | 45.9±8.8 | 66.6±12.9 | 4.8±1.3 | 45.0±12.3 | 52.0±11.9 |
| FFCMv2(Rd=1) | 18.2±12.6 | 62.7±12.6 | 79.1±10.2 | 13.0±11.4 | 48.9±9.7 | 60.4±9.6 |
| FFCMv2(Rd=10) | 4.5±0.7 | 49.4±19.3 | 56.7±16.7 | 4.9±0.6 | 44.6±9.1 | 49.9±7.3 |
| $k$-FED($k'$=5) | 73.1±14.3 | 65.6±15.2 | 68.8±12.8 | 52.7±10.8 | 43.4±6.3 | 47.5±8.6 |
| FeCA | **3.6**±0.7 | **23.6**±8.8 | **33.2**±6.7 | **4.7**±0.7 | **24.5**±7.2 | **31.5**±5.8 |
| Centralized | | 25.4±19.1 | | | 20.1±10.0 | |

*Remark* 6.1. Note that Table 1 suggests that FeCA (and some other federated clustering methods) can even outperform the centralized $k$-means. From the perspective of federated learning, this may look odd.

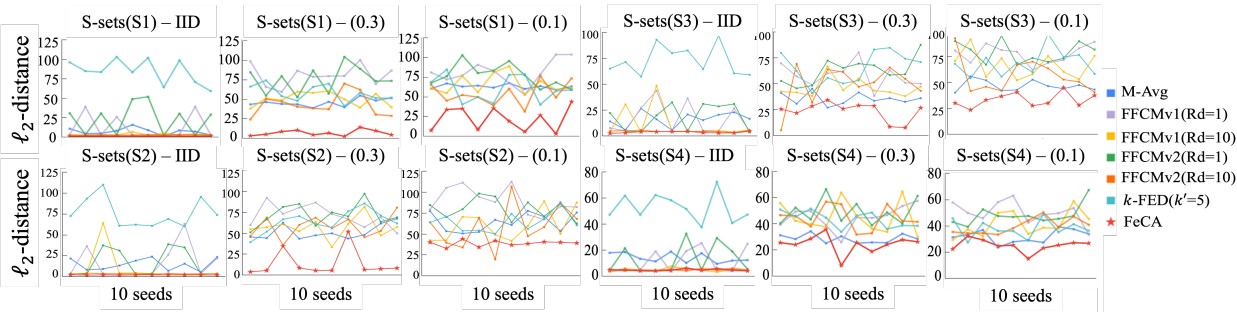

Figure 3: **Illustrations of $\ell_2$-distance results in Table 1 with 10 random seeds.**

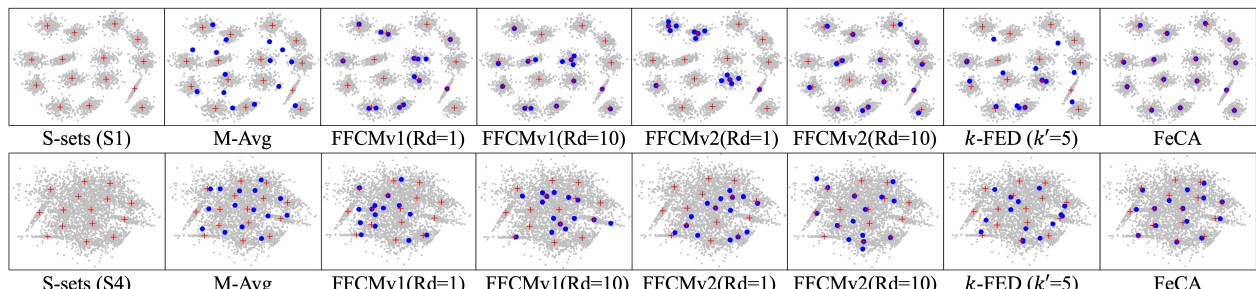

Figure 4: **Visualizations of S-sets (S1&S4) and recovered centroids by different methods.** Results are showcased under the Dirichlet(0.3) data sample scenario. Blue dots represent recovered centroids, and red crosses indicate the ground truth centers.

But it makes perfect sense from the local solution point of view of $k$-means. Solving centralized $k$-means likely leads to a local solution with suboptimal performance. However, with multiple clients independently solving $k$-means, their solutions together have a higher chance to collaboratively recover the global solution. To explore this advantage of FeCA, we conduct experiments on the impact of varying numbers of clients on the synthetic dataset, as detailed in Appendix D.

**On frozen features.** We evaluate FeCA on features extracted from pre-trained neural networks using real datasets – CIFAR10/100 Krizhevsky et al. (2009). And the frozen features are generated from the ImageNet pre-trained ResNet-50 He et al. (2016) model. For $k$-FED algorithm, we select $k' = 3$ for CIFAR10 and $k' = 10$ for CIFAR100, following the suggestion $k' \leq \sqrt{k^*}$ from their paper. In Table 2, we present the average Purity and NMI calculated from three random runs. Our algorithm FeCA demonstrates robust performance across different data sample scenarios. And it outperforms all baseline models in most cases.

Under the extreme non-IID scenario – Dirichlet(0.1), the $k$-FED algorithm tends to outperform others. This is because $k$-FED is designed for high heterogeneity, assuming each client possesses data from $k' \leq \sqrt{k^*}$ true clusters. Without considering local solution structures in $k$-means, $k$-FED relies on an accurate selection of $k'$ to reduce the chance of occurrence of spurious centroids in scenarios of high heterogeneity, as illustrated in Figure 1(right). This strategy makes its performance highly sensitive to the choice of $k'$, and it tends to decrease rapidly in less heterogeneous cases. This situation highlights the importance of addressing local solutions in federated clustering.

## 6.2 DeepFeCA Evaluation

We validate DeepFeCA on the CIFAR10/100 and the Tiny-ImageNet dataset with $64 \times 64$ resolution. We randomly initialize a ResNet-18 model and train it for 150 rounds, with all 10 clients fully participating in the process. Each round we train 5 local epochs for clients' models with batch size 128. We modified the official implementation of DeepCluster-v2 Caron et al. (2020) and followed their training details.

Table 2: **Purity and NMI on CIFAR-10/100.** We present mean results from three random runs under four data sample scenarios. For $k$-FED method, we select $k'$=3 on CIFAR-10 and $k'$=10 on CIFAR-100.

| | CIFAR-10 | | | | | | | | CIFAR-100 | | | | | | | |
| | Purity | | | | NMI | | | | Purity | | | | NMI | | | |
| Method | IID | (1.0) | (0.3) | (0.1) | IID | (1.0) | (0.3) | (0.1) | IID | (1.0) | (0.3) | (0.1) | IID | (1.0) | (0.3) | (0.1) |
|---|---|---|---|---|---|---|---|---|---|---|---|---|---|---|---|---|
| M-Avg | 0.48 | 0.45 | 0.41 | 0.39 | 0.43 | 0.41 | 0.35 | 0.34 | 0.20 | 0.19 | 0.19 | 0.18 | 0.37 | 0.37 | 0.36 | 0.36 |
| FFCMv1(Rd=1) | 0.24 | 0.32 | 0.37 | 0.36 | 0.33 | 0.37 | 0.38 | 0.36 | 0.06 | 0.07 | 0.07 | 0.07 | 0.24 | 0.25 | 0.24 | 0.25 |
| FFCMv1(Rd=10) | 0.20 | 0.20 | 0.23 | 0.24 | 0.31 | 0.31 | 0.32 | 0.31 | 0.03 | 0.03 | 0.04 | 0.03 | 0.13 | 0.13 | 0.15 | 0.14 |
| FFCMv2(Rd=1) | 0.32 | 0.44 | 0.53 | 0.56 | 0.27 | 0.43 | 0.48 | 0.50 | 0.10 | 0.11 | 0.11 | 0.12 | 0.27 | 0.30 | 0.30 | 0.32 |
| FFCMv2(Rd=10) | 0.32 | 0.43 | 0.53 | 0.56 | 0.27 | 0.43 | 0.48 | 0.50 | 0.10 | 0.10 | 0.10 | 0.12 | 0.26 | 0.28 | 0.29 | 0.31 |
| $k$-FED | 0.28 | 0.41 | 0.47 | 0.51 | 0.30 | 0.39 | 0.47 | **0.52** | 0.10 | 0.22 | 0.34 | **0.48** | 0.27 | 0.37 | 0.49 | **0.62** |
| FECA | **0.61** | **0.60** | **0.58** | **0.57** | **0.53** | **0.51** | **0.49** | 0.50 | **0.37** | **0.37** | **0.37** | 0.34 | **0.49** | **0.49** | **0.49** | 0.47 |
| Centralized | 0.64 | | | | 0.53 | | | | 0.39 | | | | 0.50 | | | |

The test accuracy is reported using linear evaluation, with the mean results of three runs presented in Table 3. Since each round of FFCM requires one communication between the server and clients, we only perform FFCM(Rd=1) given the computation resource constraint. And we set $k' = 20$ for $k$-FED on Tiny-ImageNet. We first confirm that the centralized training of DEEPCLUSTER-v2 reaches reasonable accuracy on three datasets. Also, we see that the performance in federated framework drops sharply compared to centralized learning, showing the challenges of learning deep representation on decentralized data. One possible reason could be the limited performance of $k$-means clustering results on noisy features, especially in the early rounds, leading to unreliable pseudo labels in the following supervised training step.

However, our method shows encouraging improvements. The DEEPFECA outperforms the baselines significantly and consistently across all settings on three datasets. From experiments, we demonstrate: (1) it is challenging for applications of DEEPCLUSTER-v2 in federated settings to match its centralized performance; (2) it is possible for current baselines to learn meaningful features in federated representation learning; (3) the proposed DEEPFECA serves as a strong baseline, which outperforms current algorithms by a notable gain.

Table 3: **Top-1 test accuracy (%) with linear evaluation.** We present mean results from 3 runs on three datasets.

| Accuracy (%) | CIFAR-10 | | | CIFAR-100 | | | Tiny-ImageNet | | |
| Method | IID | (0.3) | (0.1) | IID | (0.3) | (0.1) | IID | (0.3) | (0.1) |
|---|---|---|---|---|---|---|---|---|---|
| M-Avg | 53.6 | 48.2 | 45.1 | 25.2 | 24.8 | 23.6 | 16.4 | 15.1 | 13.5 |
| FFCMv1(Rd=1) | 44.6 | 48.5 | 45.5 | 22.5 | 25.0 | 25.9 | 12.1 | 16.9 | 12.5 |
| FFCMv2(Rd=1) | 47.7 | 50.1 | 49.1 | 21.9 | 22.9 | 22.9 | 14.9 | 14.1 | 14.0 |
| $k$-FED($k'$=20) | 46.8 | 50.0 | 47.1 | 27.5 | 28.9 | 29.7 | 15.3 | 18.9 | 23.4 |
| DEEPFECA | **55.3** | **63.7** | **59.9** | **35.3** | **34.9** | **35.2** | **26.7** | **26.4** | **26.1** |
| Centralized | 80.9 | | | 50.1 | | | 36.6 | | |

## 7 Conclusions and Future Works

We investigate federated clustering, an important yet under-explored area in federated unsupervised learning, and propose a one-shot algorithm, FeCA, by leveraging structures of local solutions in $k$-means. We also adopt FeCA for representation learning and propose DeepFeCA. Through comprehensive experiments on benchmark datasets, both FeCA and DeepFeCA demonstrate superior performance and robustness, outperforming established baselines across various settings.

**Towards other challenging settings.** Throughout the whole paper, we consider either the infinite sample scenario (for theory) or the large sample scenario (for experiments), where a considerable large data sample size is still required for the local solution to have the desired structure. This corresponds to the cross-silo federated learning as introduced in the review paper Kairouz et al. (2021), where the number of clients is limited but sufficient data is available on each client. The other cross-device federated learning setting, where limited data is available on each client, can be adapted to tentatively mimic the cross-silo setting. One could group all the clients into a few groups to guarantee sufficient data samples in each group, and then apply FedAvg McMahan et al. (2017) or other federated algorithm over the clients within each group, and at last deploy our federated $k$-means algorithm on solutions returned by different groups.

### Acknowledgments

J. Xu and Y. Zhang acknowledge support from the Department of Electrical and Computer Engineering at Rutgers University. H.-Y. Chen and W.-L. Chao are supported in part by grants from the National Science Foundation (IIS-2107077 and OAC-2112606) and Cisco Research.

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

## A  Proof of Theorem 4.1

In this section, we prove Theorem 4.1. Under the Stochastic Ball Model and high Signal-to-Noise Ratios (SNR) condition, we will demonstrate: (1) all the clients returned one/many-fit-one centroids corresponding to the same ground truth center are bounded within a ball of some radius (determined by Algorithm 3); (2) for the final output centroids $\mathcal{C}^* = \{c_1^*, \ldots, c_k^*\}$ from Algorithm 1, the distance between any centroid $c_s^* \in \mathcal{C}^*, s \in [k]$ and its corresponding true center is upper bounded. Specifically, the proof is composed of the following three steps:

- **Step 1 (Proof of the effectiveness of removing one-fit-many in Algorithm 2)**: On the client end, we prove the effectiveness of removing all one-fit-many centroids by Algorithm 2;

- **Step 2 (Proof of the effectiveness of assigning radius in Algorithm 3)**: On the server end, we prove that the radius assigned by Algorithm 3 effectively encloses all the centroids (returned from the clients) associated with the same true center;

- **Step 3 (Proof of Theorem 4.1)**: On the server end, under the assumption that there does not exist any one-fit-many centroid (proved in Step 1), we first prove Algorithm 5 correctly classifies all the returned centroids $\mathcal{C} = \{\mathcal{C}^{(1)}, \ldots, \mathcal{C}^{(M)}\}$ from $M$ clients using radii assigned by Algorithm 3, and then derive the error bound between recovered centroids and their associated ground truth centers.

For completeness, we outline some notations used in the following proof and a formal description of the Stochastic Ball Model below.

**Stochastic Ball Model and Notations.**  Let $\theta_1^*, \ldots, \theta_k^* \in \mathbb{R}^d$ represent $k$ distinct true cluster centers, and $f_s$ be the density of a distribution with mean $\theta_s^*$ for each $s \in [k]$. We assume each data point $x^{(m)} \in \mathbb{R}^d$ of client $m \in [M]$ is sampled independently and uniformly from a mixture $f$ of distributions $\{f_s\}_{s \in [k]}$, with the density

$$f(x) = \frac{1}{k} \sum_{s=1}^{k} f_s(x). \tag{7}$$

The Stochastic Ball Model is the mixture $f$ where each ball component has density

$$f_s(x) = \frac{1}{\text{Vol}(\mathbb{B}_s)} \mathbb{1}_{\mathbb{B}_s}(x), \quad s \in [k], \tag{8}$$

where $\mathbb{B}_s$ denotes a ball component centered at $\theta_s^*$ with radius $r$. In the context of the $k$-means problem, to identify a set of $k$ centroids $\mathcal{C}^{(m)} = \{c_1^{(m)}, \ldots, c_k^{(m)}\}$ on client $m$, we consider the goal as minimizing the following objective:

$$G(\mathcal{C}^{(m)}) = N \int \min_{i \in [k]} \|x^{(m)} - c_i^{(m)}\|_2^2 f(x^{(m)}) dx^{(m)} = \frac{1}{k} \sum_{s=1}^k \int \min_{i \in [k]} \|x^{(m)} - c_i^{(m)}\|_2^2 f_s(x^{(m)}) dx^{(m)}. \quad (9)$$

The above objective function represents the infinite-sample limit of the objective (1) on client $m$. In the following proof, we denote the objective $G(\mathcal{C}^{(m)})$ in (9) on client $m$ as $G^{(m)}$. Given a set of $k$ centroids $\mathcal{C}^{(m)}$, we denote the associated Voronoi set as $\{\mathcal{V}_1^{(m)}, \ldots, \mathcal{V}_k^{(m)}\}$, where $\mathcal{V}_j^{(m)}$ is the region consisting of all the points closer to $c_j^{(m)}$ than any other centroid in $\mathcal{C}^{(m)}$. Formally, for each $j \in [k]$, we define

$$\mathcal{V}_j^{(m)} = \{x : \|x - c_j^{(m)}\|_2 \le \|x - c_l^{(m)}\|_2, \forall l \neq j, l \in [k]\}. \quad (10)$$

In addition, we define the maximum and minimum pairwise separations between true centers $\{\theta_s^*\}_{s \in [k]}$ as

$$\Delta_{max} := \max_{s \neq s'} \|\theta_s^* - \theta_{s'}^*\|_2 \qquad \text{and} \qquad \Delta_{min} := \min_{s \neq s'} \|\theta_s^* - \theta_{s'}^*\|_2. \quad (11)$$

### A.1 Step 1 (Proof of the effectiveness of removing one-fit-many in Algorithm 2)

In this section, we prove the effectiveness of Algorithm 2 in eliminating all one-fit-many centroids under the Stochastic Ball Model. Specifically, on each client $m \in [M]$, after applying Lloyd's algorithm, we obtain a set of $k$ centroids $\mathcal{C}^{(m)}$. If $\mathcal{C}^{(m)}$ is a non-degenerate local minimum that is not the global optimum, then it must contain both one-fit-many and many-fit-one centroids, as discussed in subsection 3.1.

Next, the algorithm identifies a candidate one-fit-many centroid $c_i^{(m)} \in \mathcal{C}^{(m)}$ whose corresponding Voronoi set $\mathcal{V}_i^{(m)}$ contains data points with the largest standard deviation of distances to $c_i^{(m)}$. In this proof, with the infinite-sample limit, we derive the objective $G_i^{(m)}$ in Equation 3 as

$$G_i^{(m)} = \int_{\mathcal{V}_i^{(m)}} \|x - c_i^{(m)}\|_2^2 f(x) dx. \quad (12)$$

In addition, the algorithm pinpoints two candidate many-fit-one centroids $c_p^{(m)}$ and $c_q^{(m)}$ from $\mathcal{C}^{(m)}$, characterized by the minimal pairwise distance. Then we tentatively merge the respective Voronoi sets $\mathcal{V}_p^{(m)}$ and $\mathcal{V}_q^{(m)}$ to form a new region $\mathcal{D}_j^{(m)}$ and subsequently obtain a corresponding centroid $c_j^{(m)}$. The objective $G_j^{(m)}$ in Equation 4 is then calculated as

$$G_j^{(m)} = \int_{\mathcal{D}_j^{(m)}} \|x - c_j^{(m)}\|_2^2 f(x) dx, \quad \text{where} \quad \mathcal{D}_j^{(m)} = \mathcal{V}_p^{(m)} \cup \mathcal{V}_q^{(m)}. \quad (13)$$

To test if these candidate centroids $c_i^{(m)}$ and $\{c_p^{(m)}, c_q^{(m)}\}$ are one-fit-many and many-fit-one centroids, respectively, Algorithm 2 compares $G_i^{(m)}$ and $G_j^{(m)}$. If $G_i^{(m)} \ge G_j^{(m)}$, then $c_i^{(m)}$ is confirmed as a one-fit-many centroid to be removed, and $\{c_p^{(m)}, c_q^{(m)}\}$ are many-fit-one centroids to be kept. This is proved in the following Lemma A.1.

**Lemma A.1.** *Under the Stochastic Ball Model, for some constants $\lambda \ge 3$ and $\eta \ge 5$, if*

$$\Delta_{max} \ge 4\lambda^2 k^4 r \qquad and \qquad \Delta_{min} \ge 10\eta\lambda k^2 \sqrt{r\Delta_{max}}, \quad (14)$$

*then Algorithm 2 eliminates all the one-fit-many centroids in a local minimizer $\mathcal{C}^{(m)}$ on client $m$.*

*Proof.* If $\mathcal{C}^{(m)}$ is a non-degenerate local minimum that is not globally optimal on client $m$, then it must contain both one-fit-many and many-fit-one centroids. Without loss of generality, assume that $c_i^{(m)} \in \mathcal{C}^{(m)}$

is associated with multiple true centers $\theta_1^*, \ldots, \theta_t^*$, where $t \geq 2$. Additionally, let $\{c_p^{(m)}, c_q^{(m)}\} \in \mathcal{C}^{(m)}$ (potentially along with other centroids) are associated with the same true center $\theta_{t+1}^*$. Then the objective $G^{(m)}$ of $\mathcal{C}^{(m)}$ is

$$G^{(m)} = G_i^{(m)} + T_1 + B, \quad \text{where} \quad T_1 = \int_{\mathcal{V}_p^{(m)} \cup \mathcal{V}_q^{(m)}} \min\{\|x - c_p^{(m)}\|_2^2, \|x - c_q^{(m)}\|_2^2\} f(x) dx, \tag{15}$$

and $G_i^{(m)}$ is defined in (12). $B$ denotes the objective value contributed by the Voronoi set other than $\{\mathcal{V}_i^{(m)}, \mathcal{V}_p^{(m)}, \mathcal{V}_q^{(m)}\}$.

We construct a hypothetical solution $\mathcal{C}_h^{(m)}$ by: (1) merging Voronoi sets $\mathcal{V}_p^{(m)}$ and $\mathcal{V}_q^{(m)}$ into a new region $\mathcal{D}_j^{(m)}$ with the centroid $c_j^{(m)}$; (2) dividing the Voronoi set $\mathcal{V}_i^{(m)}$ into two regions $\mathcal{D}_{i_1}^{(m)}$ and $\mathcal{D}_{i_2}^{(m)}$, with new centroids $c_{i_1}^{(m)}$ and $c_{i_2}^{(m)}$ respectively:

$$\mathcal{D}_{i_1}^{(m)} = \{x \in \mathcal{V}_i^{(m)} : \|x - c_{i_1}^{(m)}\|_2 \leq \|x - c_{i_2}^{(m)}\|_2\}, \quad \mathcal{D}_{i_2}^{(m)} = \{x \in \mathcal{V}_i^{(m)} : \|x - c_{i_2}^{(m)}\|_2 \leq \|x - c_{i_1}^{(m)}\|_2\};$$

(3) keeping the remaining centroids in $\mathcal{C}^{(m)}$. Thus, we have

$$\mathcal{C}_h^{(m)} = \mathcal{C}^{(m)} \setminus \{c_i^{(m)}, c_p^{(m)}, c_q^{(m)}\} \cup \{c_{i_1}^{(m)}, c_{i_2}^{(m)}, c_j^{(m)}\}. \tag{16}$$

The objective $G_h^{(m)}$ for this hypothetical solution $\mathcal{C}_h^{(m)}$ is

$$G_h^{(m)} = G_j^{(m)} + T_2 + B, \quad \text{where} \quad T_2 = \int_{\mathcal{V}_i^{(m)}} \min\{\|x - c_{i_1}^{(m)}\|_2^2, \|x - c_{i_2}^{(m)}\|_2^2\} f(x) dx, \tag{17}$$

and $G_j^{(m)}$ is defined in (13). By selecting centroids $c_{i_1}^{(m)} = c_i^{(m)}$ and $c_{i_2}^{(m)} = \arg\max_{x \in \mathcal{V}_i^{(m)}} \|x - c_i^{(m)}\|$ and applying Lemma A.2, we have

$$G^{(m)} - G_h^{(m)} \geq \frac{\Delta_{min}^2}{36k}. \tag{18}$$

This inequality implies that $\mathcal{C}^{(m)}$ is a local solution with a suboptimal objective value $G^{(m)}$. In Algorithm 2, instead of comparing $G^{(m)}$ and $G_h^{(m)}$, we evaluate $G_i^{(m)}$ and $G_j^{(m)}$ to determine whether $\mathcal{C}^{(m)}$ is a local solution. The difference between objective values $G^{(m)}$ and $G_h^{(m)}$ is

$$G^{(m)} - G_h^{(m)} = G_i^{(m)} - G_j^{(m)} - (T_2 - T_1). \tag{19}$$

Following the selection of centroids $c_{i_1}^{(m)}$ and $c_{i_2}^{(m)}$ as above, we have the proved claim that $\|c_{i_1}^{(m)} - c_{i_2}^{(m)}\| \geq \frac{\Delta_{min}}{2} - r$ from the proof of Lemma A.2 in Hong et al. (2022). For the term $T_2$, it follows that

$$T_2 = \int_{\mathcal{D}_{i_1}^{(m)}} \|x - c_{i_1}^{(m)}\|_2^2 f(x) dx + \int_{\mathcal{D}_{i_2}^{(m)}} \|x - c_{i_2}^{(m)}\|_2^2 f(x) dx \tag{20}$$

$$\geq \int_{\mathcal{D}_{i_2}^{(m)}} \left(\|c_{i_1}^{(m)} - c_{i_2}^{(m)}\|_2 - \|x - c_{i_1}^{(m)}\|_2\right)^2 f(x) dx \tag{21}$$

$$\geq \int_{\mathcal{D}_{i_2}^{(m)}} \left(\frac{\Delta_{min}}{2} - r - 2r\right)^2 f(x) dx \tag{22}$$

$$\geq \frac{1}{k} \left(\frac{\Delta_{min}}{2} - 3r\right)^2 \tag{23}$$

$$\geq \frac{1}{k} \left(\frac{2\Delta_{min}}{5}\right)^2 \tag{24}$$

Equation (20) follows from $\mathcal{V}_i^{(m)} = \mathcal{D}_{i_1}^{(m)} \cup \mathcal{D}_{i_2}^{(m)}$. Inequality (21) uses the triangle inequality, and (22) follows from the choice of $c_{i_1}^{(m)}$ and $c_{i_2}^{(m)}$. Inequality (23) stems from the ball component's volume being $\frac{1}{k}$. Inequality (24) follows the claim that $r \leq \frac{\Delta_{min}}{20\eta\lambda^2 k^4} \leq \frac{\Delta_{min}}{30}$, which is derived from the separation assumption in (14).

For the term $T_1$, we have

$$T_1 = \int_{\mathcal{V}_p^{(m)}} \|x - c_p^{(m)}\|_2^2 f(x)dx + \int_{\mathcal{V}_q^{(m)}} \|x - c_q^{(m)}\|_2^2 f(x)dx \tag{25}$$

$$\leq \int_{\mathcal{V}_p^{(m)}} \left( \|x - \theta_{t+1}^*\|_2 + \|\theta_{t+1}^* - c_p^{(m)}\|_2 \right)^2 f(x)dx + \int_{\mathcal{V}_q^{(m)}} \left( \|x - \theta_{t+1}^*\|_2 + \|\theta_{t+1}^* - c_q^{(m)}\|_2 \right)^2 f(x)dx \tag{26}$$

$$\leq \int_{\mathcal{V}_p^{(m)}} \left( r + 8\lambda k^2 \sqrt{r\Delta_{max}} \right)^2 f(x)dx + \int_{\mathcal{V}_q^{(m)}} \left( r + 8\lambda k^2 \sqrt{r\Delta_{max}} \right)^2 f(x)dx \tag{27}$$

$$= \int_{\mathcal{D}_j^{(m)}} \left( r + 8\lambda k^2 \sqrt{r\Delta_{max}} \right)^2 f(x)dx \leq \frac{1}{k} \left( r + \frac{4\Delta_{min}}{5\eta} \right)^2 \leq \frac{1}{k} \left( \frac{\Delta_{min}}{3} \right)^2 \tag{28}$$

Equation (25) follows from the definition of the term $T_1$. Inequality (26) follows from the triangle inequality. Inequality (27) utilizes the error bound from Theorem 1 (many/one-fit-one association) in Qian et al. (2021), with each ball component's radius as $r$. Inequality (28) is based on the volume of each ball component being $\frac{1}{k}$, the proved claim $r \leq \frac{\Delta_{min}}{30}$, and the assumption that $\eta \geq 5$.

Combining the above inequalities, we have $T_2 - T_1 \geq 0$. Thus, the equation (19) can be derived as

$$G_i^{(m)} - G_j^{(m)} = G^{(m)} - G_h^{(m)} + (T_2 - T_1) \geq G^{(m)} - G_h^{(m)} \geq \frac{\Delta_{min}^2}{36k} \geq 0. \tag{29}$$

Therefore, if $G_i^{(m)} \geq G_j^{(m)}$, it implies that $\mathcal{C}^{(m)}$ is a local solution with a suboptimal objective value $G^{(m)}$. Subsequently, by comparing $G_i^{(m)}$ and $G_j^{(m)}$ for each candidate centroid $c_i^{(m)}$, Algorithm 2 can effectively eliminates all one-fit-many centroids from $\mathcal{C}^{(m)}$. $\qquad\square$

**Lemma A.2.** *Under the Stochastic Ball Model, for some constants $\lambda \geq 3$ and $\eta \geq 5$, if*

$$\Delta_{max} \geq 4\lambda^2 k^4 r \qquad and \qquad \Delta_{min} \geq 10\eta\lambda k^2 \sqrt{r\Delta_{max}}, \tag{30}$$

*then when $c_{i_1}^{(m)} = c_i^{(m)}$ and $c_{i_2}^{(m)} = \arg\max_{x \in \mathcal{V}_i^{(m)}} \|x - c_i^{(m)}\|$, the following holds:*

$$G^{(m)} - G_h^{(m)} \geq \frac{\Delta_{min}^2}{36k}. \tag{31}$$

*Proof.* The difference between objective values of the solution $C^{(m)}$ and the hypothetical solution $C_h^{(m)}$ is

$$G^{(m)} - G_h^{(m)} = (G_i^{(m)} - T_2) - (G_j^{(m)} - T_1). \tag{32}$$

Lemma A.2 in Hong et al. (2022) establishes that by choosing centroids $c_{i_1}^{(m)} = c_i^{(m)}$ and $c_{i_2}^{(m)} = \arg\max_{x \in \mathcal{V}_i^{(m)}} \|x - c_i^{(m)}\|$, we have

$$G_i^{(m)} - T_2 \geq \frac{\Delta_{min}^2}{18k}, \quad \text{and} \quad G_j^{(m)} - T_1 \leq \frac{4r^2}{k}, \tag{33}$$

which follows from the volumes of the ball components under the Stochastic Ball Model, each equating to $\frac{1}{k}$. Then, we derive

$$G^{(m)} - G_h^{(m)} \geq \frac{\Delta_{min}^2}{18k} - \frac{4r^2}{k} \geq \frac{\Delta_{min}^2}{36k}. \tag{34}$$

The second inequality in (34) follows the claim that $r \leq \frac{\Delta_{min}}{20\eta\lambda^2 k^4} \leq \frac{\Delta_{min}}{30}$, which is derived from the separation assumption in (30).

$\qquad\square$

## A.2 Step 2 (Proof of the radius assignment in Algorithm 3)

In this section, we present a theoretical analysis demonstrating the effectiveness of the radius assignment in Algorithm 3, ensuring coverage of all centroids associated with one true center. On one hand, following the removal of all one-fit-many centroids by Algorithm 2 (proved in step 1), all returned centroids $\mathcal{C} = \{\mathcal{C}^{(1)}, \ldots, \mathcal{C}^{(M)}\}$ on the server end are concentrated around true centers $\{\theta_s^*\}_{s \in [k]}$. Thus, this is equivalently another clustering problem on centroids $\mathcal{C}$ with (extremely) high SNR separation condition. Let $\{\mathcal{S}_1^*, \ldots, \mathcal{S}_k^*\}$ be the ground truth clustering sets of all returned centroids $\mathcal{C}$, where for each $s \in [k]$, centroids within $\mathcal{S}_s^*$ are all associated with the one true center $\theta_s^*$. Algorithm 5 classifies these returned centroids $\mathcal{C}$ into $k$ sets $\{\mathcal{S}_1, \ldots, \mathcal{S}_k\}$, using the radius in $\mathcal{R} = \{\mathcal{R}^{(1)}, \ldots, \mathcal{R}^{(M)}\}$ determined by Algorithm 3.

On each client $m \in [M]$, Algorithm 3 first generates a new set of centroids $\tilde{\mathcal{C}}^{(m)}$ by discarding any potential many-fit-one centroid from $\mathcal{C}^{(m)}$. It then identifies the minimal pairwise distance $\tilde{\Delta}_{min}^{(m)}$ in $\tilde{\mathcal{C}}^{(m)}$ aiming to approximate $\Delta_{min}$, formulated as:

$$\tilde{\Delta}_{min}^{(m)} = \min_{\tilde{c}_i, \tilde{c}_j \in \tilde{\mathcal{C}}^{(m)}, i \neq j} \|\tilde{c}_i - \tilde{c}_j\|_2. \tag{35}$$

Subsequently, Algorithm 3 calculates a uniform radius $r^{(m)} = \frac{1}{2}\tilde{\Delta}_{min}^{(m)}$, and assigns it to every centroid in $\mathcal{C}^{(m)}$. These centroid-radius pairs are then sent to the server.

**Lemma A.3.** *Under the Stochastic Ball Model, for some constant $\lambda \geq 3$ and $\eta \geq 5$, if*

$$\Delta_{max} \geq 4\lambda^2 k^4 r \qquad and \qquad \Delta_{min} \geq 10\eta\lambda k^2 \sqrt{r\Delta_{max}}, \tag{36}$$

*then for centroids in $\mathcal{S}_s^*$ which are associated with the true center $\theta_s^*, s \in [k]$, we have the following inequality holds on all clients $m \in [M]$:*

$$\max_{c_i, c_j \in \mathcal{S}_s^*, i \neq j} \|c_j - c_i\|_2 \leq \frac{1}{2}\tilde{\Delta}_{min}^{(m)}, \quad \forall s \in [k]. \tag{37}$$

*Proof.* For centroids in $\mathcal{S}_s^*$ which are associated with one true center $\theta_s^*, s \in [k]$, we upper bound their maximum pairwise distance using the triangle inequality:

$$\max_{c_i, c_j \in \mathcal{S}_s^*, i \neq j} \|c_j - c_i\|_2 \leq \max_{c_i, c_j \in \mathcal{S}_s^*, i \neq j} \left(\|c_j - \theta_s^*\|_2 + \|\theta_s^* - c_i\|_2\right)$$
$$\leq 2\max_{c_i \in \mathcal{S}_s^*} \|c_i - \theta_s^*\|_2. \tag{38}$$

Under the Stochastic Ball Model with radius $r$, we apply the error bound from Theorem 1 (many/one-fit-one association) in Qian et al. (2021) to the above inequality, and for some constant $\lambda \geq 3$ we have

$$\max_{c_i, c_j \in \mathcal{S}_s^*, i \neq j} \|c_j - c_i\|_2 \leq 2\max_{c_i \in \mathcal{S}_s^*} \|c_i - \theta_s^*\|_2 \leq 16\lambda k^2 \sqrt{r\Delta_{max}}. \tag{39}$$

By combining the above inequality and the assumption $\Delta_{min} \geq 10\eta\lambda k^2 \sqrt{r\Delta_{max}}$ in (36), we obtain

$$\max_{c_i, c_j \in \mathcal{S}_s^*, i \neq j} \|c_j - c_i\|_2 \leq \frac{8}{5\eta}\Delta_{min}, \tag{40}$$

where the constant $\eta \geq 5$. Next, we derive the approximation error between $\tilde{\Delta}_{min}^{(m)}$ and $\Delta_{min}$ as:

$$|\tilde{\Delta}_{min}^{(m)} - \Delta_{min}| \leq \max_{c_s \in \mathcal{S}_s^*} \|c_s - \theta_s^*\|_2 + \max_{c_{s'} \in \mathcal{S}_{s'}^*} \|c_{s'} - \theta_{s'}^*\|_2, \qquad s \in [k], s' \in [k], s \neq s'$$
$$\leq 2\max_{c_s \in \mathcal{S}_s^*} \|c_s - \theta_s^*\|_2, \qquad s \in [k]. \tag{41}$$

Given that all clients follow the same mixture distributions under the Stochastic Ball Model, the above inequality holds for all clients $m \in [M]$. Similar to inequality (38), we again utilize the error bound from Theorem 1 (many/one-fit-one association) in Qian et al. (2021), and obtain:

$$|\tilde{\Delta}_{min}^{(m)} - \Delta_{min}| \leq \frac{8}{5\eta}\Delta_{min}, \qquad \forall m \in [M]. \tag{42}$$

Reorganizing the terms in inequality (42) gives

$$\frac{5\eta}{5\eta + 8}\tilde{\Delta}_{min}^{(m)} \leq \Delta_{min} \leq \frac{5\eta}{5\eta - 8}\tilde{\Delta}_{min}^{(m)}, \qquad \forall m \in [M]. \tag{43}$$

Then by combining inequalities (40) and (43), for each $s \in [k]$, we obtain

$$\max_{c_i, c_j \in \mathcal{S}_s^*, i \neq j} \|c_j - c_i\|_2 \leq \frac{8}{5\eta - 8}\tilde{\Delta}_{min}^{(m)} \leq \frac{1}{2}\tilde{\Delta}_{min}^{(m)}, \qquad \forall m \in [M], \tag{44}$$

where the last inequality follows from the assumption that $\eta \geq 5$. $\qquad\square$

## A.3 Proof of Theorem 4.1

In this section, we complete the proof of our main theorem by demonstrating: (1) Algorithm 5 correctly classifies all returned centroids in alignment with their corresponding true centers, utilizing the radius assigned by Algorithm 3; (2) we derive the error bound between the final output centroids $\mathcal{C}^*$ from Algorithm 1 and the corresponding true centers.

*Proof.* Let cluster labels be $s = 1, \ldots, k$. During the grouping process of all returned centroids $\mathcal{C}$ on the server end, Algorithm 5 first selects a centroid in $\mathcal{C}$ with the largest radius. Without loss of generality, we assume that this selected centroid $c_s \in \mathcal{C}$ is returned by the client $m$ and associated with the true center $\theta_s^*$. Thus, this centroid belongs to the ground truth clustering set as $c_s \in \mathcal{S}_s^*$ and its assigned radius is $r_s = \frac{1}{2}\tilde{\Delta}_{min}^{(m)}$. Algorithm 5 then groups the centroids located within the ball centered at $c_s$ with radius $r_s$, resulting in the formation of the grouped cluster $\mathcal{S}_s = \{c : c \in \mathcal{C}, \|c - c_s\| \leq r_s\}$.

On one hand, Lemma A.3 implies that for each $s \in [k]$, the maximum pairwise distance between centroids in $\mathcal{S}_s^*$ is bounded by $\frac{1}{2}\tilde{\Delta}_{min}^{(m)}$ for any client $m$. Consequently, $\mathcal{S}_s^* \in \mathcal{S}_s$ can be readily inferred based on the definition of $\mathcal{S}_s$.

On the other hand, for other centroids $c_{s'} \in \mathcal{S}_{s'}^*, s' \in [k], s \neq s'$, we have

$$\begin{aligned}
\|c_{s'} - c_s\|_2 &\geq \|\theta_{s'}^* - \theta_s^*\|_2 - \|c_{s'} - \theta_{s'}^*\|_2 - \|c_s - \theta_s^*\|_2 \\
&\geq \Delta_{min} - \|c_{s'} - \theta_{s'}^*\|_2 - \|c_s - \theta_s^*\|_2.
\end{aligned} \tag{45}$$

Utilizing the error bound from Theorem 1 (many/one-fit-one association) in Qian et al. (2021) gives

$$\|c_{s'} - c_s\|_2 \geq \Delta_{min} - 16\lambda k^2\sqrt{r\Delta_{max}} \geq \Delta_{min} - \frac{8}{5\eta}\Delta_{min}, \tag{46}$$

where the last step follows from the assumption $\Delta_{min} \geq 10\eta\lambda k^2\sqrt{r\Delta_{max}}$. Applying the lower bound in (43) to the above inequality, for any client $m \in [M]$, it follows that

$$\|c_{s'} - c_s\|_2 \geq \frac{5\eta - 8}{5\eta + 8}\tilde{\Delta}_{min}^{(m)} > \frac{1}{2}\tilde{\Delta}_{min}^{(m)}, \tag{47}$$

where the constant $\eta \geq 5$. Thus, following the definition of $\mathcal{S}_s$, the above inequality implies that centroids $c_{s'} \in \mathcal{S}_{s'}^*, s \neq s'$ do not belong to $\mathcal{S}_s$. Combining the proved claim that $\mathcal{S}_s^* \in \mathcal{S}_s$, this suggests that $\mathcal{S}_s = \mathcal{S}_s^*, s \in [k]$, up to a permutation of cluster labels. This further implies that Algorithm 5 correctly classifies all returned centroids in $\mathcal{C}$ according to their associated true centers.

For each $s \in [k]$, Algorithm 5 computes the mean of $\mathcal{S}_s$, denoted as $c_s^* = \text{mean}(\mathcal{S}_s)$. The collection of these mean centroids, $\mathcal{C}^* = \{c_1^*, \dots, c_k^*\}$, constitutes the final set of centroids output by Algorithm 1. Then the proximity of $c_s^*$ to its associated true center $\theta_s^*$ can be bounded as:

$$\|c_s^* - \theta_s^*\|_2 \leq \max_{c \in \mathcal{S}_s} \|c - \theta_s^*\|_2 \leq \max_{c \in \mathcal{S}_s^*} \|c - \theta_s^*\|_2, \quad \forall s \in [k] \tag{48}$$

$$\leq 8\lambda k^2 \sqrt{r\Delta_{max}} \leq \frac{4}{5\eta}\Delta_{min}, \tag{49}$$

for some constants $\eta \geq 5$. The inequality (48) follows from the proved statement $\mathcal{S}_s = \mathcal{S}_s^*$, $s \in [k]$. The inequality (49) first utilizes the error bound from Theorem 1 (many/one-fit-one association) in Qian et al. (2021), followed by the application of the separation assumption $\Delta_{min} \geq 10\eta\lambda k^2 \sqrt{r\Delta_{max}}$. Thus, any output centroid $c_s^* \in \mathcal{C}^*$ from Algorithm 1 is close to some true center $\theta_{s'}^*$ as:

$$\|c_s^* - \theta_{s'}^*\|_2 \leq \frac{4}{5\eta}\Delta_{min}, \tag{50}$$

thereby proving Theorem 4.1.

$\square$

## B   Evaluation on the radius assigned by Algorithm 4 (*Empirical*).

This section evaluates the radius produced by the empirical algorithm variant, Algorithm 4. While the proof of our main theorem characterizes the performance of the Algorithm 3, it is important to note that most real-world scenarios do not satisfy a homogeneous data sample assumption. Consequently, we have implemented an empirical procedure that assigns a unique radius to each returned centroid. In this context, it is also not necessary to remove many-fit-one centroids on clients, as these centroids typically concentrate around true centers and will be grouped together via the aggregation algorithm on the server. This empirical algorithm variant is specifically designed to adapt our main algorithm FECA for more general scenarios.

In this section, we empirically assess the radius assigned by Algorithm 4 under both IID and non-IID scenarios, demonstrating its effectiveness for the proceeding aggregation step in Algorithm 5. Specifically, our objective is to empirically show that utilizing the selected centroid-radius pair $(c_s, r_s), s \in [k]$ (assigned by Algorithm 4), Algorithm 5 can effectively group all returned centroids corresponding to the same true center $\theta_s^*$ on the server end. Recall that we denote a set of returned centroids associated with one true center $\theta_s^*$ as $\mathcal{S}_s^*$.

Essentially, we aim to validate that the distance between any centroid $c \in \mathcal{S}_s^*$ to $c_s$ is bounded by the assigned radius $r_s$. Our goal is thus formulated as follows:

$$\max_{c_i \in \mathcal{S}_s^*} \|c_i - c_s\|_2 \leq r_s. \tag{51}$$

The left side of the above inequality indicates the maximum distance between any two centroids in $\mathcal{S}_s^*$, and it can be further elaborated as

$$\max_{c_i \in \mathcal{S}_s^*} \|c_i - c_s\|_2 \leq \max_{c_s \in \mathcal{S}_s^*} (\|c_i - \theta_s^*\|_2 + \|\theta_s^* - c_s\|_2) \leq 2 \max_{c_i \in \mathcal{S}_s^*} \|c_i - \theta_s^*\|_2. \tag{52}$$

Then our goal in (51) can be reformulated as

$$\max_{c_i \in \mathcal{S}_s^*} \frac{\|c_i - \theta_s^*\|_2}{r_s} \leq \frac{1}{2}. \tag{53}$$

Next, we present empirical results on the synthetic dataset, S-sets (S1), with known ground truth centers $\{\theta_s^*\}_{s \in [k]}$. These results demonstrate that the inequality (53) holds across both IID and non-IID cases. For this purpose, we define a new parameter $\sigma$ as

$$\sigma_i := \frac{\|c_i - \theta_s^*\|_2}{r_s}, \quad c_i \in \mathcal{S}_s^* \quad s \in [k]. \tag{54}$$

This parameter $\sigma_i$ represents the distance between the returned centroid $c_i$ and its fitted true center $\theta_s^*$ scaled by the radius $r_s$. Our empirical results demonstrate that values of $\sigma_i$ remain below 0.5 for all returned centroids in $\mathcal{C} = \{\mathcal{C}^{(1)}, \ldots, \mathcal{C}^{(M)}\}$, in accordance with our goal inequality (53).

**Results.** Figure 5 illustrates the evaluation of $\sigma_i$, as determined using the radius assigned by Algorithm 4, in varied inhomogeneous settings on S-sets (S1). This figure presents $\sigma_i$ values for all returned centroids across three random runs, categorized according to their respective true centers in different colors. Result consistently indicate that $\sigma_i$ values stay below 0.5, thereby empirically substantiating the validity of the inequality (53) in our analysis. Consequently, it demonstrates the efficacy of aggregating centroids using the radius assigned by Algorithm 4 on the server end.

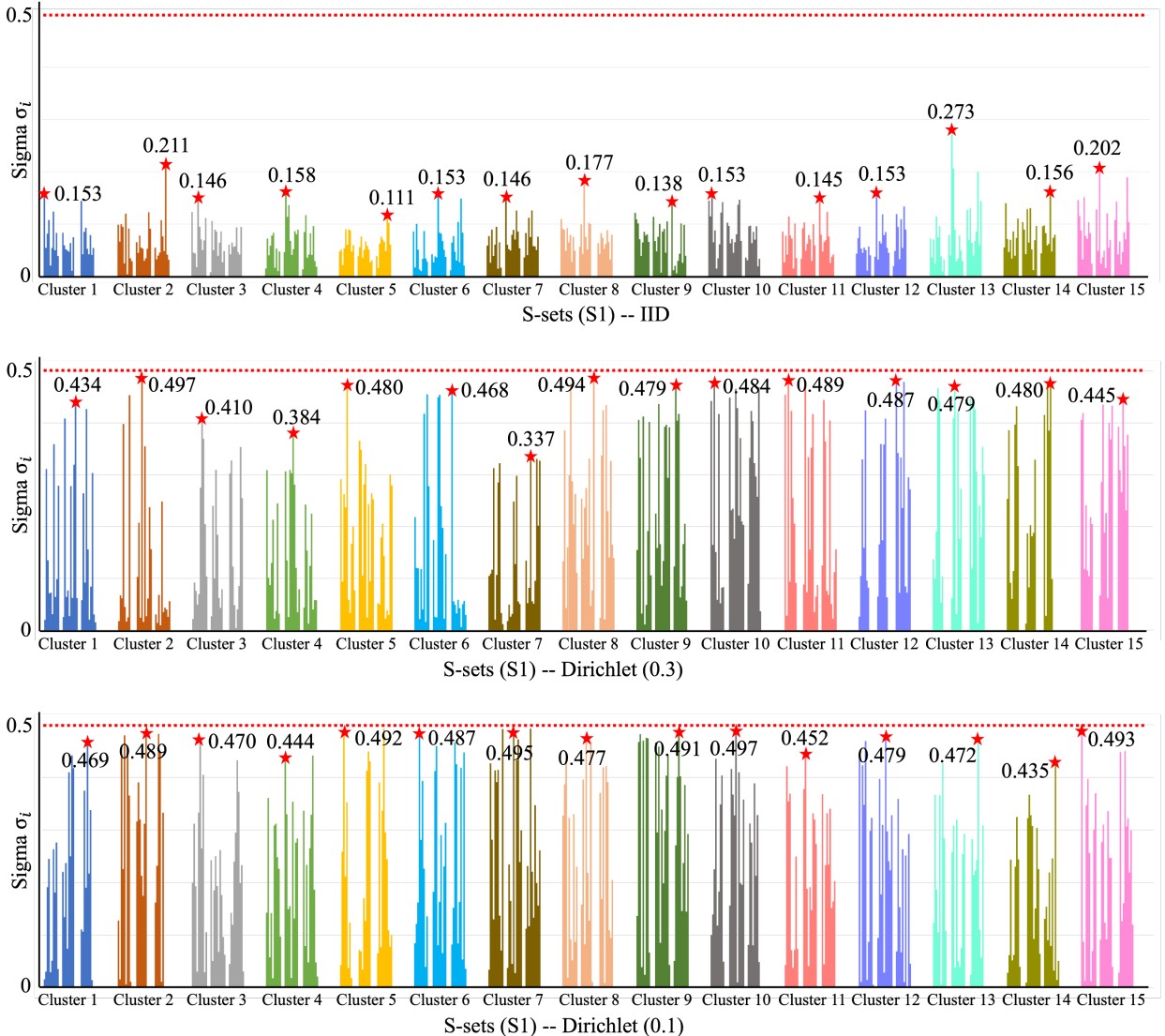

Figure 5: **Evaluation of $\sigma$ on S-sets (S1) across three data sample scenarios.** $\sigma_i$ for $k$ clusters is represented in different colors. The values of $\sigma_i$ for all returned centroids $c_i$ are reported over 3 random runs, with the red star marking the maximum $\sigma_i$ observed in three runs. *A $\sigma_i$ value below 0.5 indicates that, the server effectively groups centroid $c_i$ utilizing the radius $r_s$ assigned by Algorithm 4.*

We note that the number of returned centroids associated with each true center may vary. It is because we selectively remove one-fit-many centroids on the client side, while it is possible for many-fit-one centroids to be present. In some extreme non-IID cases, assuming a client only contains a few secluded data points from one true cluster but they all far deviate from the true center, it may occur that a returned centroid is not covered by the radius. Then it will be considered noisy and discarded by Algorithm 5. Concretely, the recovered centroids of this cluster will be contributed by returned centroids from other clients.

## C  Supplementary experiments on the synthetic dataset

This section presents additional experimental results on the synthetic dataset S-sets Fränti & Sieranoja (2018). The S-sets comprise four sets: S1, S2, S3, and S4, each consisting of 15 Gaussian clusters in 2-dimensional data with varying degrees of overlap, specifically $9\%, 22\%, 41\%$, and $44\%$. For the visualization of S-sets, refer to Figure 6 from their paper Fränti & Sieranoja (2018).

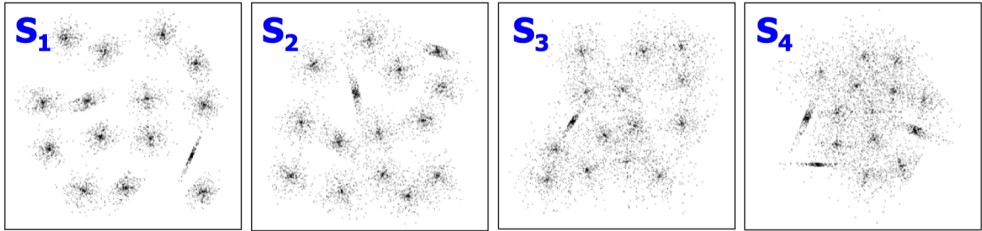

Figure 6: **Visualizations of S-sets.**

### C.1  Evaluations on the clustering assignments

This section shifts focus to the evaluation of clustering assignments on the synthetic dataset S-sets, diverging from the analysis of recovered centroids. While Table 1 in the paper assesses the $\ell_2$-distance between recovered centroids and known ground truth centers, we herein present the average results of Purity and

Table 4: **Purity and NMI on S-sets under three data sample scenarios.**

| Purity | S-sets (S1) | | | S-sets (S2) | | | S-sets (S3) | | | S-sets (S4) | | |
|---|---|---|---|---|---|---|---|---|---|---|---|---|
| Method | IID | (0.3) | (0.1) | IID | (0.3) | (0.1) | IID | (0.3) | (0.1) | IID | (0.3) | (0.1) |
| M-Avg | 0.99 | 0.79 | 0.69 | 0.93 | 0.74 | 0.68 | 0.80 | 0.66 | 0.61 | 0.75 | 0.64 | 0.57 |
| FFCMv1 (Rd=1) | 0.97 | 0.69 | 0.66 | 0.95 | 0.68 | 0.58 | 0.83 | 0.60 | 0.54 | 0.79 | 0.60 | 0.53 |
| FFCMv1 (Rd=10) | 0.98 | 0.78 | 0.68 | 0.96 | 0.74 | 0.71 | 0.85 | 0.66 | 0.57 | 0.80 | 0.61 | 0.55 |
| FFCMv2 (Rd=1) | 0.95 | 0.69 | 0.66 | 0.95 | 0.63 | 0.61 | 0.83 | 0.58 | 0.52 | 0.78 | 0.57 | 0.54 |
| FFCMv2 (Rd=10) | 0.99 | 0.80 | 0.76 | 0.97 | 0.70 | 0.72 | 0.86 | 0.65 | 0.61 | 0.80 | 0.60 | 0.59 |
| $k$-FED ($k'$=5) | 0.62 | 0.76 | 0.75 | 0.60 | 0.71 | 0.71 | 0.55 | 0.61 | 0.60 | 0.55 | 0.62 | 0.58 |
| FᴇCA | **0.99** | **0.98** | **0.96** | **0.97** | **0.95** | **0.90** | **0.86** | **0.80** | **0.78** | **0.80** | **0.73** | **0.65** |
| Centralized | 0.97 | | | 0.96 | | | 0.83 | | | 0.77 | | |
| **NMI** | S-sets (S1) | | | S-sets (S2) | | | S-sets (S3) | | | S-sets (S4) | | |
| Method | IID | (0.3) | (0.1) | IID | (0.3) | (0.1) | IID | (0.3) | (0.1) | IID | (0.3) | (0.1) |
| M-Avg | 0.98 | 0.85 | 0.79 | 0.92 | 0.80 | 0.78 | 0.77 | 0.71 | 0.69 | 0.70 | 0.65 | 0.62 |
| FFCMv1 (Rd=1) | 0.98 | 0.80 | 0.78 | 0.93 | 0.77 | 0.71 | 0.78 | 0.67 | 0.64 | 0.72 | 0.63 | 0.60 |
| FFCMv1 (Rd=10) | 0.98 | 0.85 | 0.80 | 0.94 | 0.80 | 0.78 | 0.79 | 0.71 | 0.66 | 0.72 | 0.64 | 0.61 |
| FFCMv2 (Rd=1) | 0.97 | 0.80 | 0.78 | 0.93 | 0.75 | 0.72 | 0.78 | 0.67 | 0.63 | 0.71 | 0.62 | 0.60 |
| FFCMv2 (Rd=10) | 0.99 | 0.87 | 0.85 | 0.95 | 0.78 | 0.79 | 0.79 | 0.70 | 0.68 | 0.72 | 0.63 | 0.63 |
| $k$-FED ($k'$=5) | 0.80 | 0.84 | 0.83 | 0.75 | 0.80 | 0.80 | 0.66 | 0.69 | 0.68 | 0.62 | 0.65 | 0.63 |
| FᴇCA | **0.99** | **0.96** | **0.95** | **0.95** | **0.94** | **0.90** | **0.80** | **0.77** | **0.75** | **0.72** | **0.69** | **0.66** |
| Centralized | 0.98 | | | 0.94 | | | 0.79 | | | 0.71 | | |

NMI across 10 random runs in Table 4 under three different data sample scenarios. The findings consistently demonstrate that our algorithm FeCA surpasses all baseline algorithms in performance across every tested scenario, underlining its effectiveness in federated clustering tasks.

## C.2 More visualizations of recovered centroids by different methods on S-sets

To further demonstrate the superior performance of our algorithm, we present more visualizations corresponding to the results detailed in Table 1 for S-sets(S2) and S-sets(S3). Figure 7 displays the centroids recovered by various federated clustering algorithms under the non-IID condition – Dirichlet(0.3). Our algorithm's ability to resolve and leverage the structures of local solutions enables it to outperform other baseline methods that fail to address these critical aspects, especially in challenging non-IID settings. This emphasizes the critical role of resolving local solutions for enhanced algorithmic performance.

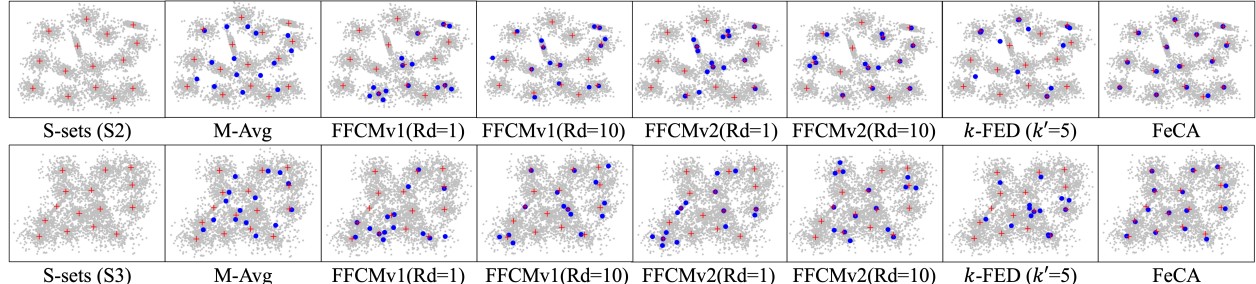

Figure 7: **Visualizations of S-sets (S2&S3) and recovered centroids by different methods.** Results are showcased under the Dirichlet(0.3) data sample scenario. Blue dots represent recovered centroids, and red crosses indicate the ground truth centers.

## C.3 Ablation study on eliminating one-fit-many centroids in Algorithm 2

Removing one-fit-many centroids in Algorithm 2 plays a crucial role in enhancing the algorithm's performance. These centroids are typically far from any true centers. By eliminating one-fit-many centroids at the client end, we effectively prevent the transmission of these problematic centroids to the server. It significantly simplifies the task of Algorithm 5 on the server side, which involves grouping received centroids close to the same true center.

In this section, we conduct an ablation study on eliminating one-fit-many centroids in Algorithm 2. In the following experiments, one-fit-many centroids are not removed on clients and then sent to the server. We present mean square errors between recovered centroids and ground truth centers in Table 5. The comparative results clearly demonstrate a performance degradation when these centroids are not removed, underscoring the significance of eliminating the one-fit-many step in Algorithm 2.

Table 5: **Mean square errors between recovered centroids and true centers on S-sets(S1) under IID data sample scenario.** Values of MSE are scaled by $10^6$. We report mean results from 10 random runs.

| Method | MSE $\times 10^6$ |
| --- | --- |
| FeCA (not removing one-fit-many centroids) | 12977.2 |
| FeCA (removing one-fit-many centroids) | **6.7** |

**Not enough output centroids.** Not removing one-fit-many centroids can lead to a scenario where the number of reconstructed centroids is less than $k$. This occurs because the cluster of one-fit-many centroid typically contains data points from multiple true clusters, resulting in a significantly larger radius than that assigned to the true cluster. Consequently, the server may prioritize these centroids with large radii during the grouping process, forming a large group erroneously containing centroids associated with different true centers. In Figure 8, we provide visualizations of reconstructed centroids without removing one-fit-many,

demonstrating a notable decrease in performance. This emphasizes the necessity of their removal in our algorithm.

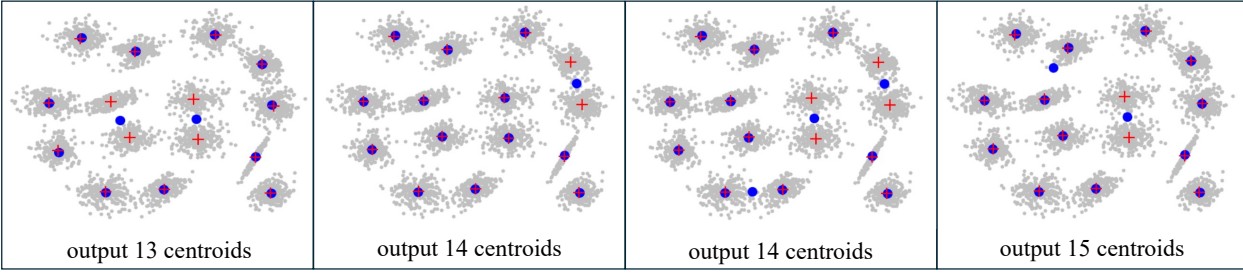

| output 13 centroids | output 14 centroids | output 14 centroids | output 15 centroids |

Figure 8: **Visualizations of recovered centroids by FeCA (not removing one-fit-many centroids).** Results are showcased under the IID condition on S-sets(S1) with $k^* = 15$. Blue dots represent recovered centroids, and red crosses indicate the ground truth centers.

### C.4 Comparison between Algorithm 3 (theoretical) and Algorithm 4 (empirical)

Algorithm 3 (theoretical) is designed for theoretical analysis only under a strict setup, specifically the Stochastic Ball Model assumption. This assumption allows for the straightforward identification and removal of many-fit-one centroids. However, in practical scenarios, eliminating many-fit-ones is challenging and unnecessary, as they often carry crucial information about the global solution.

In this section, we explore the applicability of Algorithm 3 (theoretical) beyond its constraints by conducting experiments on S-sets under various heterogeneous conditions. Table 6 presents results, revealing that the performance of Algorithm 3 is suboptimal when assumptions of Stochastic Ball Model are not met, particularly in non-IID cases. This suboptimal performance is due to its reliance on specific assumptions for identifying many-fit-one centroids. Consequently, in practical scenarios, this approach may erroneously eliminate true centroids, leading to less effective outcomes.

Table 6: $\ell_2$-**distance (mean±std) between recovered centroids and true centers on S-sets(S1) under different data sample scenarios.** Values of $\ell_2$-distance are scaled by $10^4$.

| Method | IID | (0.3) | (0.1) |
|---|---|---|---|
| FeCA (theoretical) | 1.1 ±0.1 | 31.8 ± 23.8 | 58.9 ± 15.1 |
| FeCA (empirical) | **1.0**±0.1 | **6.8**±3.7 | **22.3**±15.6 |

### C.5 Tuning $k'$ for $k$-FED on the synthetic dataset

In this section, we provide the results of experiments conducted to select $k'$ for $k$-FED Dennis et al. (2021) on the synthetic dataset. Given that all four subsets of S-sets have the same number of true clusters $k^* = 15$, here we utilize S-sets(S1) for tuning $k'$, which has the largest degree of separation. This choice is based on the separation condition mentioned in their paper. We present the $\ell_2$-distance between recovered centroids and true centers for $k'$ values ranging from 2 to 15. And results (mean±std) from 10 random runs are reported in Table 7. Additionally, we evaluate clustering assignments generated from the recovered centroids using Purity and NMI metrics. The mean results from 10 random runs are included in Table 7.

Considering $k$-FED is designed for heterogeneous cases, we adopt a Dirichlet(0.3) data sample scenario in our experiments. It is also important to note that when $k' = k^*$, the aggregation step in the $k$-FED algorithm essentially becomes redundant, and the recovered centroids are equivalent to the set of centroids returned by one randomly chosen client.

Table 7: **$\ell_2$-distance (mean±std) between recovered centroids and true centers, Purity, and NMI on S-sets(S1) under Dirichlet(0.3) data sample scenario.** Values of $\ell_2$-distance are scaled by $10^4$.

| $k$-FED | $k' = 2$ | $k' = 3$ | $k' = 4$ | $k' = 5$ | $k' = 6$ | $k' = 7$ | $k' = 8$ |
|---|---|---|---|---|---|---|---|
| $\ell_2$-distance | 63.3±7.0 | 65.9±11.9 | 60.8±16.8 | **58.1±10.7** | 60.3±15.0 | 59.4±11.1 | 62.5±11.0 |
| Purity | 0.71 | 0.72 | 0.79 | **0.79** | 0.78 | 0.79 | 0.76 |
| NMI | 0.83 | 0.83 | **0.89** | 0.88 | 0.87 | 0.88 | 0.87 |

| $k$-FED | $k' = 9$ | $k' = 10$ | $k' = 11$ | $k' = 12$ | $k' = 13$ | $k' = 14$ | $k' = 15$ |
|---|---|---|---|---|---|---|---|
| $\ell_2$-distance | 77.3±11.2 | 69.4±13.8 | 77.7±15.2 | 76.4±10.7 | 72.1±16.1 | 69.8±15.7 | 67.4±9.5 |
| Purity | 0.72 | 0.74 | 0.71 | 0.72 | 0.74 | 0.75 | 0.78 |
| NMI | 0.83 | 0.84 | 0.83 | 0.85 | 0.85 | 0.86 | 0.85 |

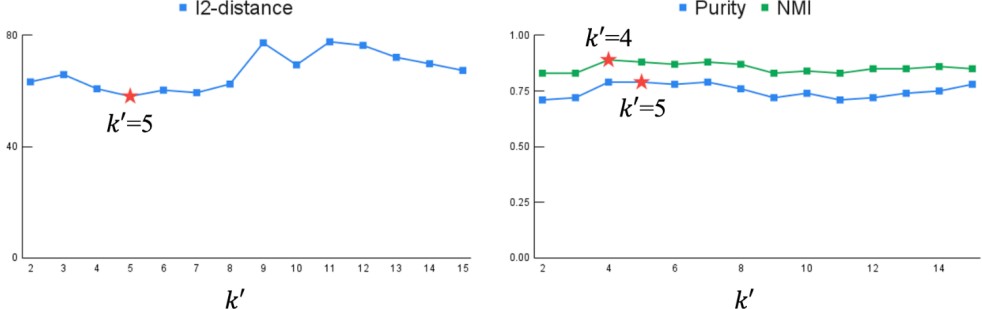

Figure 9: **Illustrations of $\ell_2$-distance means, Purity and NMI detailed in Table 7.**

## C.6 Presence of local solutions in $k$-means

In this section, we perform centralized $k$-means on 50%, 75%, and 100% of data from S-sets. In Figure 10, we illustrate objective values calculated as Equation 1 using 10 random seeds. As depicted, Lloyd's algorithm frequently converges to local solutions with objective values significantly larger than that of global solutions. This empirical result demonstrates the presence of local solutions with poor performance is independent of the data sample size. Also, it highlights the necessity of our algorithm, which is specifically designed to address and resolve these suboptimal local solutions.

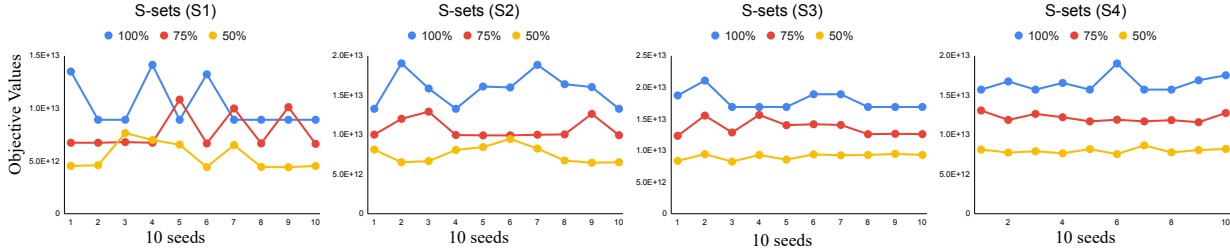

Figure 10: **Objective values of centralized $k$-means using $100\%$, $75\%$, and $50\%$ of data from S-sets.**

To better understand the structures of local solutions, we visualize some local solutions of centralized $k$-means on S-sets, shown in Figure 11. This visualization reveals that despite variations in data separations, spurious local solutions exhibit structures as discussed in subsection 3.1, containing both one-fit-many and many-fit-one centroids. This emphasizes the necessity of the steps in our algorithm that discard one-fit-many and aggregate many-fit-one centroids.

**Almost-empty cases.** As outlined in Qian et al. (2021), local solutions of k-means can be composed of one/many-fit-one, one-fit-many and almost-empty centroids. The first two have been discussed in detail

**Local Solutions of Centralized k-means on S-sets**

Figure 11: **Visualizations of structured local solutions of centralized $k$-means on S-sets.**

in subsection 3.1. Addressing almost-empty cases involves identifying centroids that are far from any true centers and its cluster is almost empty with a small measure. This typically occurs when the dataset contains isolated points that are significantly far from the true centers. It is worth noting that almost-empty cases are more theoretical than practical, with rare occurrences in empirical experiments.

However, if such a case does occur, our algorithm can handle it in the aggregation step by Algorithm 5. Centroids from almost-empty clusters can be treated as noisy data and discarded during the grouping process. Since they are distant from true centers and other received centroids, they do not contribute meaningfully to the final grouping.

## D   Discussion on varying numbers of clients

In this section, we explore the impact of varying client numbers on our federated approach. We conduct experiments by allocating a fixed dataset portion (5% randomly sampled from the entire dataset) to each client and then perform our algorithm FECA across varying numbers of clients. We first evaluate the recovered centroids by calculating $\ell_2$-distance between these centroids and the ground truth centers. Subsequently, we apply a one-step Lloyd's algorithm using the recovered centroids for initialization and then evaluate the clustering assignments by calculating Purity and NMI. Results for the S-sets (S1) are presented in Table 8. It is noteworthy that only centralized $k$-means clustering is performed when the number of clients is one. In such cases, centralized $k$-means often results in large $\ell_2$-distance due to convergence to local optima with suboptimal performance.

The findings presented in Table 8 are visually depicted in Figure 12 (left), where a trend of decreasing $\ell_2$-distance is observed as the number of clients $M$ increases. This trend indicates that collaboration among multiple clients can significantly mitigate the negative impact of local solutions. Specifically, when a client encounters a local minimum, integrating benign results from other clients can help alleviate this issue. This collaborative mechanism underscores the effectiveness of federated approaches in improving performance by leveraging the distributed nature of client contributions.

Additionally, we assess the impact of varying client numbers in a standard federated setting, particularly under IID data sample scenario for S-sets(S1). In the following experiments, with $M$ denoting the number of clients, each client is allocated $\frac{1}{M}$ of the data points from S-sets(S1). We note that centralized $k$-means is performed when $M = 1$ on the entire dataset. We evaluate the performance of our algorithm FECA by reporting the $\ell_2$-distance between recovered centroids and true centers, alongside the Purity and NMI of clustering assignments, detailed in Table 9. Moreover, Figure 12 (right) features visual representations of the $\ell_2$ distance results, emphasizing the robustness of our federated algorithm FECA across varying numbers of clients.

Table 8: $\ell_2$-**distance (mean±std) between recovered centroids and true centers, Purity, and NMI with different numbers of clients** $M$**.** Values of $\ell_2$-distance are scaled by $10^4$. Each client possesses 5% data of S-sets(S1).

| FeCA | $M=1$ | $M=2$ | $M=3$ | $M=4$ | $M=5$ | $M=6$ | $M=7$ | $M=8$ | $M=9$ | $M=10$ |
|---|---|---|---|---|---|---|---|---|---|---|
| $\ell_2$-distance | 10.7±14.3 | 3.1±0.6 | 2.6±0.4 | 2.1±0.3 | 2.0±0.3 | 1.7±0.3 | 1.8±0.2 | 1.6±0.4 | 1.4±0.2 | 1.4±0.3 |
| Purity | 0.98 | 0.99 | 0.99 | 0.99 | 0.99 | 0.99 | 0.99 | 0.99 | 0.99 | 0.99 |
| NMI | 0.98 | 0.99 | 0.99 | 0.99 | 0.99 | 0.99 | 0.99 | 0.99 | 0.99 | 0.99 |

| FeCA | $M=11$ | $M=12$ | $M=13$ | $M=14$ | $M=15$ | $M=16$ | $M=17$ | $M=18$ | $M=19$ | $M=20$ |
|---|---|---|---|---|---|---|---|---|---|---|
| $\ell_2$-distance | 1.3±0.2 | 1.4±0.3 | 1.3±0.2 | 1.4±0.2 | 1.2±0.2 | 1.2±0.1 | 1.1±0.1 | 1.1±0.1 | 1.0±0.1 | 1.0±0.1 |
| Purity | 0.99 | 0.99 | 0.99 | 0.99 | 0.99 | 0.99 | 0.99 | 0.99 | 0.99 | 0.99 |
| NMI | 0.99 | 0.99 | 0.99 | 0.99 | 0.99 | 0.99 | 0.99 | 0.99 | 0.99 | 0.99 |

Table 9: $\ell_2$-**distance (mean±std) between recovered centroids and true centers, Purity, and NMI with different numbers of clients** $M$**.** Values of $\ell_2$-distance are scaled by $10^4$. Each client possesses $\frac{1}{M}$ data of S-sets(S1).

| FeCA | $M=1$ | $M=2$ | $M=3$ | $M=4$ | $M=5$ | $M=6$ | $M=7$ | $M=8$ | $M=9$ | $M=10$ |
|---|---|---|---|---|---|---|---|---|---|---|
| $\ell_2$-distance | 14.3±18.7 | 1.1±0.3 | 1.2±0.3 | 1.1±0.2 | 1.1±0.2 | 1.2±0.2 | 1.1±0.1 | 1.1±0.1 | 1.0±0.1 | 1.0±0.1 |
| Purity | 0.97 | 0.99 | 0.99 | 0.99 | 0.99 | 0.99 | 0.99 | 0.99 | 0.99 | 0.99 |
| NMI | 0.98 | 0.99 | 0.99 | 0.99 | 0.99 | 0.99 | 0.99 | 0.99 | 0.99 | 0.99 |

| FeCA | $M=11$ | $M=12$ | $M=13$ | $M=14$ | $M=15$ | $M=16$ | $M=17$ | $M=18$ | $M=19$ | $M=20$ |
|---|---|---|---|---|---|---|---|---|---|---|
| $\ell_2$-distance | 1.1±0.1 | 1.1±0.1 | 1.1±0.2 | 1.1±0.1 | 1.1±0.1 | 1.0±0.2 | 1.1±0.2 | 1.1±0.1 | 1.1±0.1 | 1.1±0.1 |
| Purity | 0.99 | 0.99 | 0.99 | 0.99 | 0.99 | 0.99 | 0.99 | 0.99 | 0.99 | 0.99 |
| NMI | 0.99 | 0.99 | 0.99 | 0.99 | 0.99 | 0.99 | 0.99 | 0.99 | 0.99 | 0.99 |

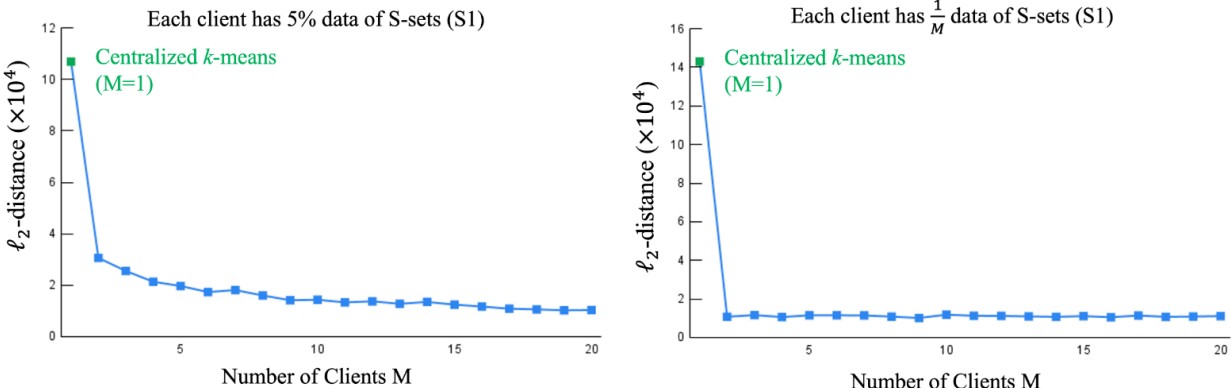

Figure 12: **Illustrations of** $\ell_2$-**distance in Table 8 and Table 9.**

# E    Discussion on varying $k$

The selection of $k$ is an important aspect of the $k$-means problem. In this section, we present supplementary experiments investigating the performance of our algorithm with varying $k$. For clarity, in this study, we use $k$ to denote the number of recovered centroids desired by our algorithm, which is also the number of output centroids by Algorithm 1. $k'$ denotes the parameter used to perform $k$-means clustering on clients in Algorithm 2, while $k^*$ indicates the number of true centers. In the following, we present experiments with varying values of $k'$ and $k$, respectively.

**Varying values of** $k'$**.** We perform FeCA on S-sets(S1) under three data sample scenarios, varying values of $k'$. We chose $k'=10$ (undershoot case) and $k'=20$ (overshoot case) with $k^*=15$. Mean square errors between recovered centroids and ground truth centers are shown in Table 10. Note that for undershoot cases,

the number of recovered centroids $k$ might be less than $k^*$. In these instances, we identify the top $k$ matching centroids with recovered centroids from $k^*$ true centers and calculate the mean square error between them. Visualizations of clustering results on different clients and FECA's recovered centroids are illustrated in the following figures.

Table 10: **Mean square errors between recovered centroids and true centers on S-sets(S1) under three data sample scenarios.** Values of MSE are scaled by $10^6$. Dirichlet(0.3) and Dirichlet(0.1) are denoted as (0.3) and (0.1), respectively. We report mean results from 10 random runs.

| Methods | $k$ | $k'$ | $k^*$ | MSE - IID | MSE - (0.3) | MSE - (0.1) |
|---|---|---|---|---|---|---|
| FECA- Varying $k'$ (undershoot case) | 15 | 10 | 15 | 34599.4 | 21055.7 | 19311.6 |
| FECA- Varying $k'$ (overshoot case) | 15 | 20 | 15 | 9.9 | 5960.4 | 8738.6 |
| FECA- Varying $k$ | 10 | 15 | 15 | 8.4 | 551.2 | 7532.7 |
| FECA | 15 | 15 | 15 | **6.7** | **308.3** | **3315.3** |

When $k' \neq k^*$, even if clients converge to the global solution, this global solution exhibits similar structures (one-fit-many and many-fit-one) to those observed in local solutions when $k' = k^*$. As shown in Figure 13, when $k' = 10$, clients' clustering results demonstrate the presence of one-fit-many, while $k' = 20$ showcases many-fit-one structures illustrated in Figure 15. Therefore, this underscores the necessity of our algorithm, as it effectively addresses such structured solutions.

In the undershoot case ($k' = 10 < k^*$), clients' clustering results might contain multiple one-fit-many but no many-fit-one centroids. This complicates the elimination of one-fit-many centroids in Algorithm 2, because identifying one-fit-many becomes challenging without many-fit-one centroids for reference. Under non-IID conditions, this might occur in extreme undershoot cases with rather small $k'$, as clients' data may concentrate on partial true clusters. As discussed in subsection C.3, if one-fit-many centroids are not eliminated on clients, the number of output centroids $k$ may be less than $k^*$, shown in Figure 13.

In contrast, in the overshoot case ($k' = 20 > k^*$), the clients' clustering results include multiple many-fit-one centroids but no one-fit-many centroids. Since our algorithm effectively addresses many-fit-one centroids in the aggregation step on the server, FECA can still accurately approximate the true centers, as demonstrated in Figure 15. This is because the elimination of many-fit-one centroids on the clients' side is unnecessary; these centroids can contribute meaningfully to the grouping process on the server side, as discussed in subsection 4.2.

**Varying values of $k$.** We perform FECA on S-sets(S1) under three data sample scenarios, selecting $k' = k^* = 15$, and $k = 10$. Mean square errors between recovered centroids and ground truth centers are reported in Table 10. When $k < k^*$, Algorithm 5 outputs the mean centroids of the top $k$ groups containing the largest number of elements. As illustrated in Figure 17 and Figure 18, with $k < k^*$, the recovered centroids can still accurately approximate partial true centers.

Clustering results on different clients when $k' = 10$.

Recovered centroids when $k = k^* = 15$.

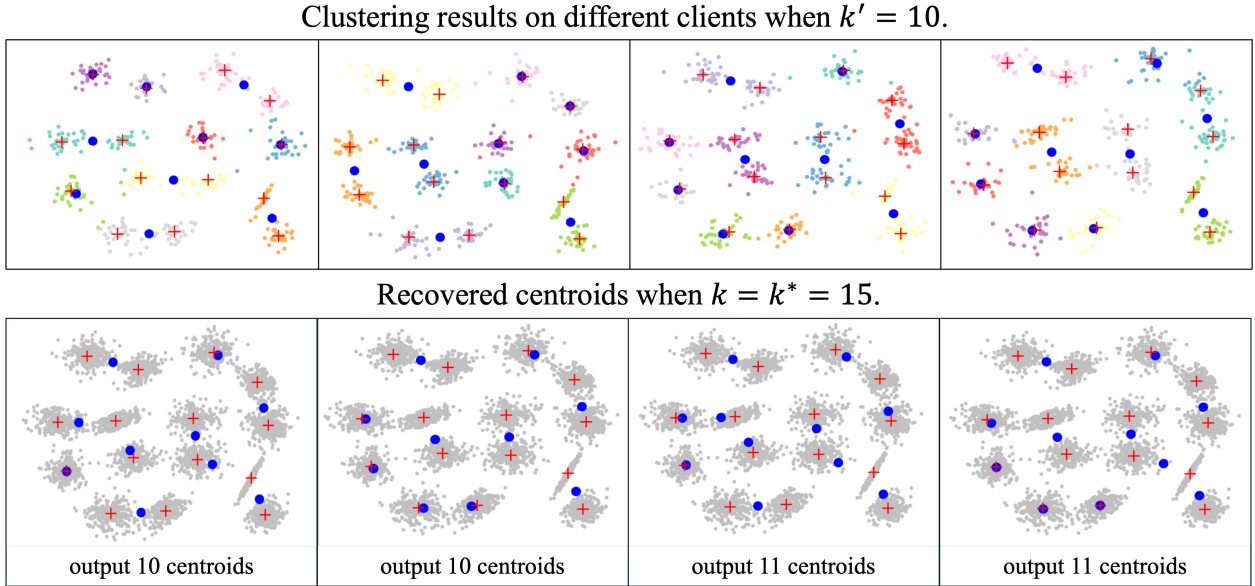

Figure 13: **Clients' clustering results and FeCA's outputs under IID condition.**

Clustering results on different clients when $k' = 10$.

Recovered centroids when $k = k^* = 15$.

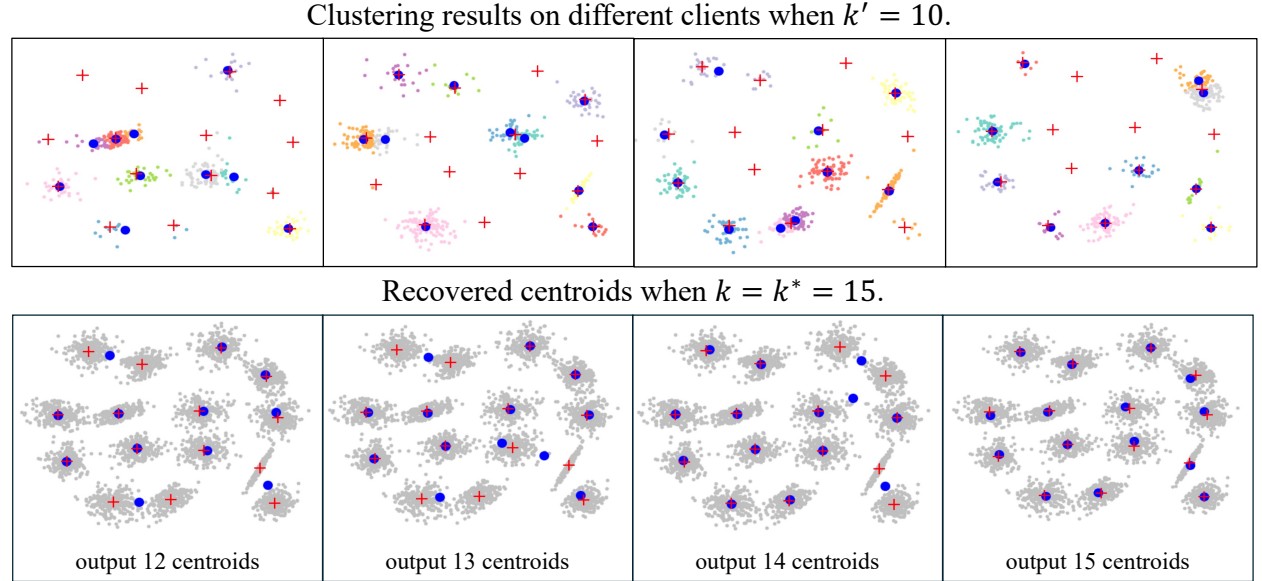

Figure 14: **Clients' clustering results and FeCA's outputs under non-IID condition - (0.3).**

Clustering results on different clients when $k' = 20$.

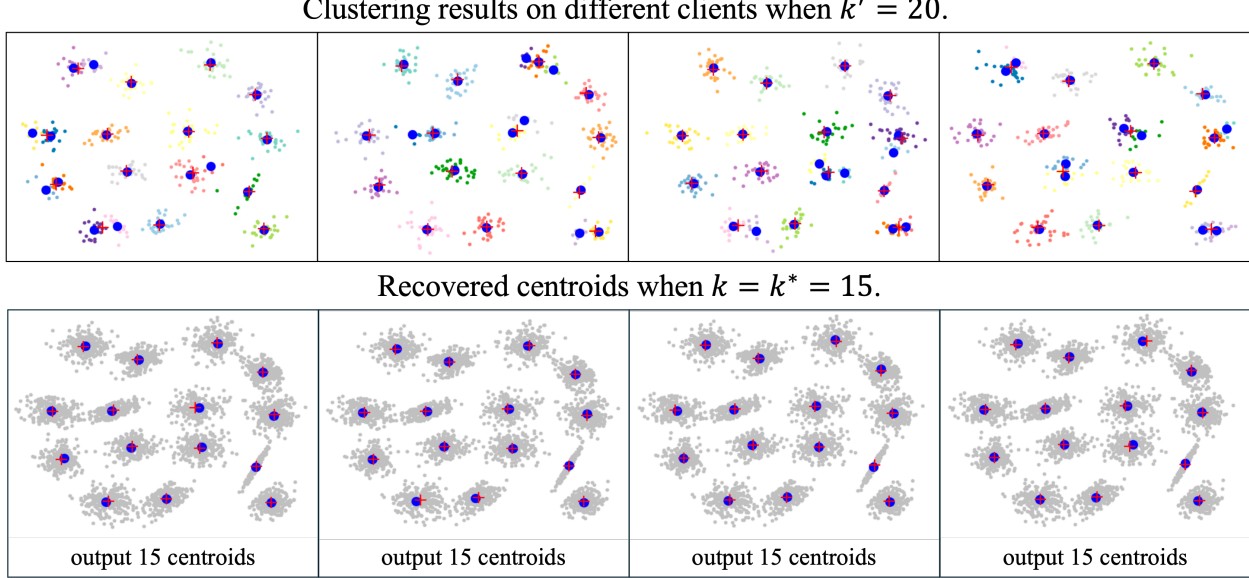

Figure 15: **Clients' clustering results and FeCA's outputs under IID condition.**

Clustering results on different clients when $k' = 20$.

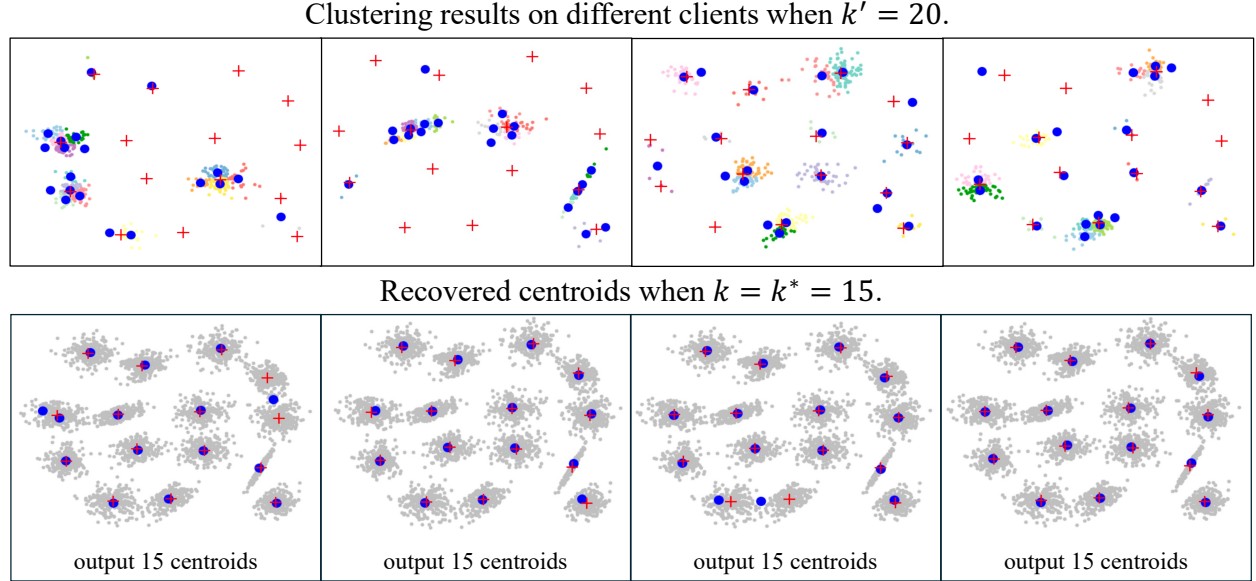

Figure 16: **Clients' clustering results and FeCA's outputs under non-IID condition - (0.3).**

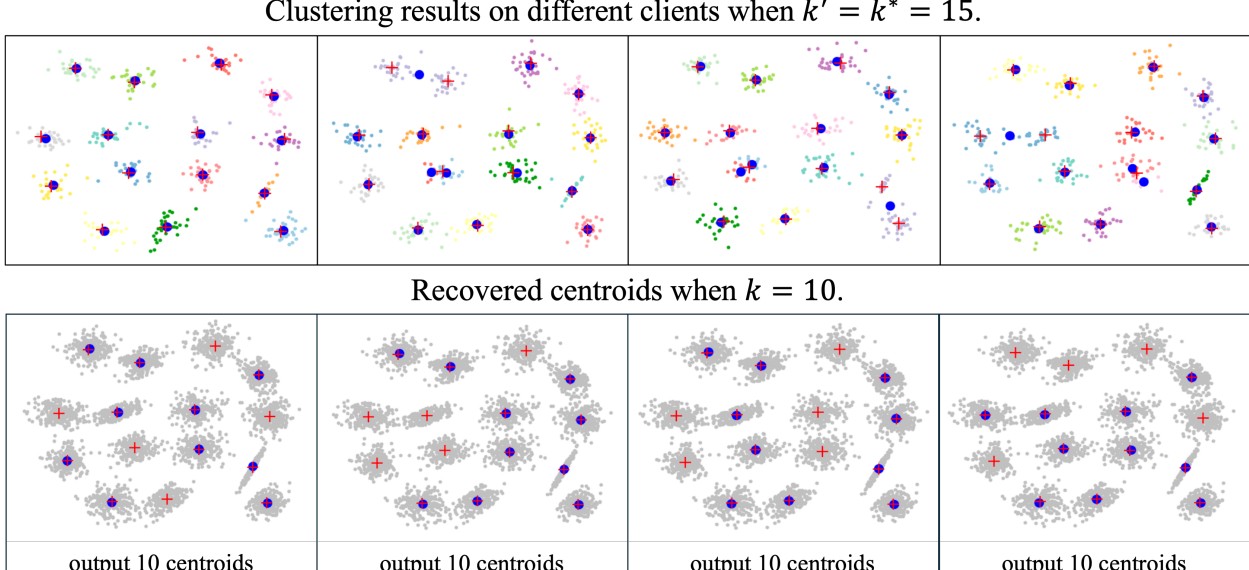

Figure 17: **Clients' clustering results and FeCA's outputs under IID condition.**

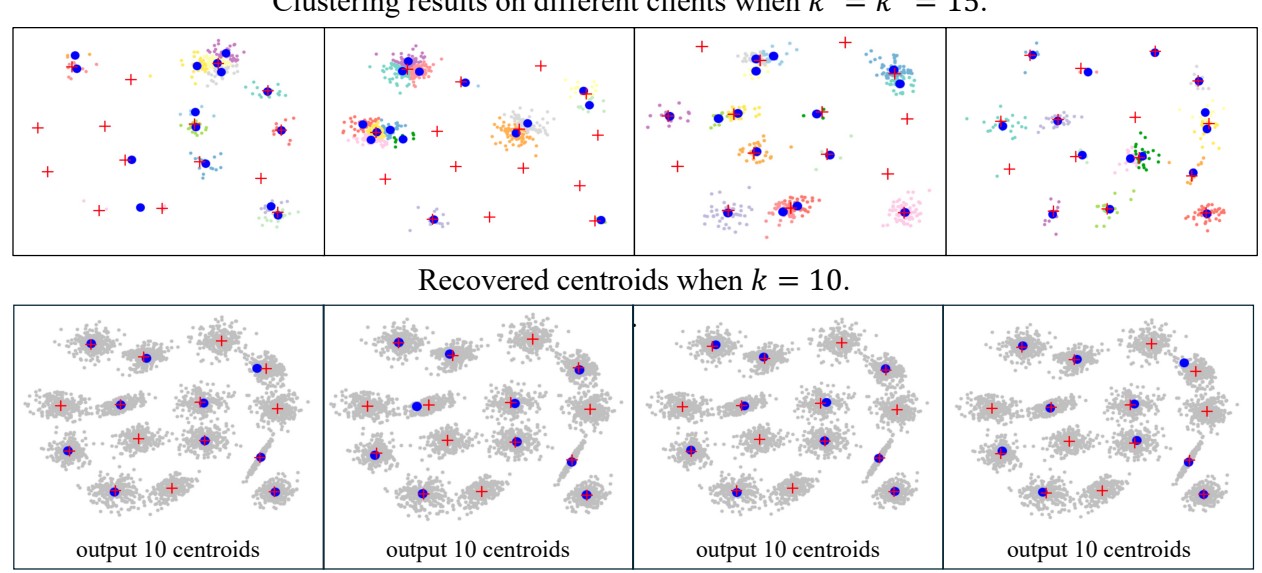

Figure 18: **Clients' clustering results and FeCA's outputs under non-IID condition - (0.3).**

