# OpenReview forum: "Jigsaw Game: Federated Clustering"
_TMLR — Accepted by TMLR_

### Review · Reviewer_puDn · 2024-04-03

**Summary Of Contributions:**

In this paper, the authors proposed a k-means algorithm for distributed environments.
They focused on the observation that the results of k-means exhibit a structure of either one-fit-many or many-fit-one, and propose a three-step algorithm.
In the first step, k-means is executed in each local environment and one-fit-many clusters are removed.
In the second step, adjacent clusters are merged in each local environment, and the radius of each cluster is estimated.
Finally, in the third stage, clusters collected from each local environment are integrated to form the final clustering result.
The authors theoretically demonstrated that under the assumption that the data follows the Stochastic Ball Model and each cluster is sufficiently separated, the proposed method FeCA can estimate true cluster centers with small errors.
Furthermore, the authors also proposed DeepFeCA, which combines FeCA with DNN to sequentially execute feature extractor learning and k-means in a distributed environment.
In experiments, the authors demonstrated that the proposed method outperforms existing methods (M-Avg, k-FED, FFCM).

**Audience:**

Yes

**Broader Impact Concerns:**

There is no ethical concern.

**Claims And Evidence:**

Yes

**Requested Changes:**

See Weakness above.

**Strengths And Weaknesses:**

### Strong aspects

#### Strength 1: Theoretical Guarantee
In Theorem 4.1, the authors theoretically demonstrated that under the assumption that the data follows the Stochastic Ball Model and each cluster is sufficiently separated, the proposed method FeCA can estimate true cluster centers with small errors.

#### Strength 2: Impact of Results
Through experiments using artificial data S-sets and CIFAR-10/100, the authors demonstrated that the proposed method FeCA outperforms existing methods (M-Avg, k-FED, FFCM).

### Weak aspects

#### Weakness: Assumption Strength
In Theorem 4.1, the authors assumed that all clusters have the same radius under the Stochastic Ball Model, and that each cluster is sufficiently separated.
Additionally, while not explicitly stated in the theorem, the proof assumes the limit as the number of data points $n \to \infty$.
Unfortunately, the paper lacks discussion on whether these assumptions are reasonable for distributed k-means.
For instance, if assuming $n \to \infty$ is valid, obtaining sufficiently good clustering results by executing k-means in each local environment might suffice.
In this case, what is the purpose of executing k-means in a distributed environment?
Moreover, the Stochastic Ball Model assumes that all clusters have the same radius, which will not be the case in most of real-world data.
It seems essential to discuss the validity of these assumptions and the robustness of the proposed method's dependency on these assumptions.

---

> ### Author Response · Authors · 2024-05-13
> **Response to Reviewer puDn**
>
> We thank the reviewer for recognizing the impact of our algorithm's results and sincerely appreciate the thoughtful review. Your comments have been carefully considered, and we have updated our manuscript accordingly.
>
> ## * Validity of the Infinite Sample Assumption.
> We kindly note that the infinite-sample assumption cannot guarantee good clustering results in the context of $k$-means. Established theoretical studies [1] have shown that when $n\rightarrow\infty$ and $k\geq 3$:
> - Local solutions in k-means can be arbitrarily worse than the global solution.
> - Randomly initialized greedy algorithms may converge to spurious local solutions with high probability.
>
> Thus, even under the infinite-sample assumption, spurious local solutions persist, underscoring the necessity of our algorithm. Our method specifically targets these local optima and leverages their benign properties in the federated clustering problem.
>
> ## * Presence of local solutions across different data sample sizes (**supplementary experiments in Appendix C.5 of the revised manuscript**).
> Despite varying data sample sizes, including infinite-sample, local solutions in k-means persist. To further demonstrate the presence of spurious local solutions, we conducted supplementary experiments with varying data sample sizes. We perform centralized k-means on 50%, 75%, and 100% of data from S-sets. Objective values from 10 random runs are presented.
>
> Results demonstrate that even as the data sample size increases, Lloyd’s algorithm still frequently converges to local solutions that are significantly worse than the global solution. This empirical evidence strongly supports the necessity of our algorithm regardless of data sample size, which is specifically designed to address and resolve these suboptimal local solutions. For detailed experiment findings, please refer to Appendix C.5.
>
>
> ## * Why we use assumptions such as Stochastic Ball Model with the same radius.
> Our analysis utilizes assumptions rooted in prior theoretical work [2], like the stochastic ball model with the same radius and well-separation assumption. Our theoretical framework builds upon the findings of [2] and thus adheres to their assumptions. While these assumptions may differ from real-world scenarios, they are commonly used in theoretical analysis for simplicity. Exploring the theoretical guarantee of our algorithm beyond these assumptions is a promising direction for future work, particularly as foundational theoretical studies[1][2] evolve.
>
> ## * Assumptions VS Practice.
>
> - **Theoretical assumptions**:
> While we use limited assumptions to simplify the analysis, it is important to clarify that our algorithm does not rely on them. We have empirically validated the effectiveness of our algorithm across a range of real-world scenarios. Furthermore, to establish theoretical support for our algorithm, we base our analysis on the findings of [2] and adhere to their assumptions.
>
> - **Practice**:
> Our algorithm’s strength is not limited by theoretical assumptions. In Section 6, we showcase its prowess across diverse datasets, including real-world ones like CIFAR10/100, and various scenarios such as non-IID split conditions. Despite these deviations, our empirical results consistently show the superiority of our algorithm over existing methods.
>
> [1] Local Maxima in the Likelihood of Gaussian Mixture Models: Structural Results and Algorithmic Consequences [[paper](https://proceedings.neurips.cc/paper/2016/file/3875115bacc48cca24ac51ee4b0e7975-Paper.pdf)]
>
> [2] Structures of Spurious Local Minima in k-means [[paper](https://arxiv.org/pdf/2002.06694.pdf)]

---

> > ### Comment · Reviewer_puDn · 2024-05-21
> > **Re: Response to Reviewer puDn**
> >
> > I would like to thank the authors for the detatiled feedback.
> >
> > Let me add a comment on the local optimality of Lloyd’s algorithm.
> > I totally agree that Lloyd’s algorithm does not guarantee global optimality.
> > However, I would also like to mention that there are some clever initialization algorithms for k-means such as k-means++ [Ref1].
> > k-means++ provides an approximation guarantee of $O(\log k)$.
> >
> > My point here is that runnning Lloyd’s algorithm with random initialization is not really a good baseline.
> > One can use k-means++ and Lloyd’s algorithm in local (with possibly infinite data) and obtain good clustering.
> >
> > [Ref1] k-means++: The Advantages of Careful Seeding

---

> > > ### Author Response · Authors · 2024-05-23
> > > **Response to Reviewer puDn**
> > >
> > > We appreciate the clarification provided and would like to address the raised point thoughtfully.
> > >
> > > ### * k-means ++ can still not guarantee global optimality.
> > > While we acknowledge that advanced initialization methods like k-means ++ can mitigate local minima issues, it is important to note that even with such methods, Lloyd's algorithm can still become trapped in local solutions, leading to suboptimal results. Moreover, k-means ++ itself is also sensitive to the initial seed or starting point. Therefore, our algorithm remains necessary for resolving structured local solutions.
> > >
> > > ### * Dealing with structured solutions is crucial for federated clustering.
> > > We would like to highlight the increased significance of addressing these structured solutions within the federated framework. In non-IID scenarios, which are common in federated settings, each client's data often originates from partial true clusters. Even if a client converges to the global solution (a good clustering result on its own data), this global solution shows similar structures, such as one-fit-many and many-fit-one, to those observed in local solutions of IID data, as illustrated in **Figure 1**. Consequently, the steps within our algorithm that discard one-fit-many and aggregate many-fit-one solutions remain crucial. This underscores why our algorithm demonstrates promising results in non-IID cases, emphasizing the effectiveness and necessity of our algorithm for resolving structured solutions in federated clustering. **We have included a detailed discussion in Appendix C.5 of the revised version.**
> > >
> > > ### * Why not choose k-means++ as baselines in experiments?
> > > Given that smart initialization methods such as k-means++ cannot effectively mitigate structured solutions in federated clustering problems under non-IID conditions, nor guarantee a global optimal solution even in IID conditions, we have chosen not to include these methods as baselines in our experiments. Our primary focus is leveraging structured solutions to address the federated clustering problem effectively.
> > >
> > > We thank you again for the valuable comments and the discussion on initialization methods within the federated clustering problem will be incorporated into the revised manuscript.

---

### Review · Reviewer_mSnm · 2024-04-24

**Summary Of Contributions:**

The paper proposes FeCA algorithm which is a one-shot k-means-type algorithm for Federated Clustering. The algorithm is based on the structure of local solutions (one-fit-many, many-fit-one [Qian et. al, 2021]). The theoretical algorithm in the first step removes all one-fit-many centroids from local solutions based on local datasets. Then it performs radius assigning process locally on each dataset, and finally the server aggregates all this information from the clients (many-fit-one local centroids and assigned radii) to obtain global set of centroids. The authors propose convergence guarantees under Stochastic Ball Model for theoretical algorithm. Besides, they propose two empirical versions that are then tested on synthetic and Deep Learning experiments.

**Audience:**

Yes

**Broader Impact Concerns:**

No ethical concerns

**Claims And Evidence:**

Yes

**Requested Changes:**

- Notation $\mathcal{D}^{(m)}$ for clusters is a little bit confusing as one can think about it as local dataset $\mathcal{D}^m$. Is it common notation to use? If not, I would encourage to use maybe $\mathcal{C}^m$ for clusters.

- Discussion on the structure of local and global solutions is needed. When is Stochastic Ball or Gaussian Mixture models are relevant and when they are not. The case of almost-empty should be also discussed. Can we observe it in some extreme practical cases?

- The discussion on why we need empirical version and why theoretical fails is needed. It would be good to have empirical comparison between then and examples when theoretical version works and fails.

- The authors should specify that the proof of theoretical version is given in infinite-n case. How large n should be?

- It would interesting to know the effect of choosing k in FeCA: when we overshoot it and undershoot. Is FeCA robust to this choice (e.g., if we overshoot k, will some centroids be close to each other meaning that k should be decreased?). I think it is one the most interesting questions for k-means-type algorithms.

Overall, I enjoyed reading this paper even though never worked on k-means algorithms. The paper's results are interesting for TMLR readers, and the requested changes would improve the paper further.

**Strengths And Weaknesses:**

Strengths:

- The paper's structure is well organized. It is intuitive and easy to follow (only few parts seem to be confusing to me; see later).
- The authors provide many empirical results: on synthetic dataset FeCA (with Dirichlet-type of heterogeneity simulations) outperforms several benchmarks in all cases significantly. Besides, provided visualizations clearly show that FeCA finds really good approximations of ground truth centroids. The algorithm seems to handle Non-IID case as well.
- The authors provide convergence guarantees under Stochastic Ball Model showing that under curtains conditions between ground truth centroids and number of clusters k.
- Proposed algorithm is FL-friendly, i.e. it requires one time connection with the server. DeepFeCA follows the style of of FedAvg that also makes it FL-applicable.
- The proofs seem to be correct to me and well written.

Weaknesses:

- The proposed algorithm hugely relies on the interaction between local and global solutions. There is a lack of discussion on this in the paper. It is not clear if this kind of structure holds for real world applications where Stochastic Ball or Gaussian Mixture models. Adding a discussion on this would improve the paper.

- Theoretical algorithm is created to remove all one-fit-many local centroids. However, then the authors claim that is not necessary to do and provide empirical version. This confuses me a lot. Especially considering the convergence guarantees for theoretical version. Why shouldn't it work in practice? I believe adding discussion on this with potential practical comparison would strength the paper. If the authors could also provide examples when theoretical version works and fails, it would be also great.

- Under Stochastic Ball model there can also be almost-empty case which is not discussed in the paper while it sounds logical to me in FL case (e.g., client has small portions of each cluster and is not able to recover any of centroids locally, so all local centroids are from true).

- There is no theoretical explanation behind good performance in practice of two empirical versions of the algorithm.

---

> ### Author Response · Authors · 2024-05-13
> **Response to Reviewer mSnm**
>
> We appreciate the reviewer for noticing the positive experimental results of our algorithm. We also appreciate the insightful suggestions provided. Your feedback has been carefully considered, and we have revised the manuscript to comprehensively address these concerns.
>
> ## * Theoretical algorithm (Alg. 3) discards **many-fit-one** centroids (not one-fit-many) as mentioned in Section 4.2, paragraph 2-3.
> We kindly refer to sentences in the original manuscript (Section 4.2, paragraph 3): “... it is both challenging and unnecessary to eliminate all many-fit-one centroids… Accordingly, we develop an empirical variant…”
>
> The aggregation process on the server (Alg. 5), is able to effectively group many-fit-one centroids. Hence, removing many-fit-one in the preceding step on clients becomes unnecessary, as they can contribute meaningfully to the grouping process.
>
> ## * Theoretical (Alg. 3) VS Empirical (Alg. 4)
> We thank the valuable comments, and a discussion of two algorithmic variants has been included in section 4.2 of the revised version.
>
> The removal of one-fit-many centroids is a necessary step occurring in Algorithm 2 before either the theoretical or empirical variants are applied. Following the elimination of one-fit-many, we develop two distinct variants to assign radii to the remaining centroids (one/many-fit-one):
> - **Theoretical variant (Alg. 3)**:
> This variant removes any potential many-fit-one centroids before assigning a unified radius to the remaining centroids. It is designed for theoretical analysis under the Stochastic Ball Model assumption. Such an assumption allows us to distinguish many-fit-one centroids easily, enabling a clearer approximation of the true cluster separation, followed by assigning the corresponding radius.
> - **Empirical variant (Alg. 4)**:
> Unlike the theoretical variant, this approach does not remove many-fit-one centroids. Instead, it assigns distinct radii to each remaining centroid, making our algorithm more effective and practical. On the one hand, it eliminates the need to handle many-fit-one centroids on clients, as they are close to true centers and can be aggregated later on the server end. On the other hand, it ensures our algorithm does not rely on any limited assumptions and remains applicable to real-world scenarios.
>
> ## * Discussion on structures of local and global solutions.
> Established theoretical work [1] characterizes local solutions with error bounds under both the Stochastic Ball Model and Gaussian Mixture Model, along with well-separated assumptions. Moreover, empirical evidence suggests that structured local solutions persist beyond these limited assumptions. **As shown in Figure 11 of the revised manuscript**, structured local solutions are observed on S-sets, which do not adhere to the theoretical assumptions. Additionally, extensive experiments on real-world datasets in Section 6 showcase our algorithm's superiority over existing methods that do not consider local solutions.
>
> **Furthermore, we conduct supplementary experiments on the presence of local solutions with varying data sample sizes, as detailed in Appendix C.5.** These experiments showcase that structured local solutions persist without the limited assumptions of datasets. A corresponding discussion has been included in Appendix C.5 of the revised version.
>
> ## * Almost-empty case.
> We appreciate the reviewer for raising this point, and a discussion on this topic has been included in Appendix C.5 of the revised manuscript.
>
> The "almost-empty case," as discussed in [1], occurs when a centroid is far from any true centers and its cluster is almost empty with a small measure. This situation occurs when the dataset contains isolated points that are significantly distant from true centers. It is worth noting that this case exists primarily in theoretical analysis and is highly unlikely to occur in practice. In all our experiments, we have never observed an almost empty case.
>
> Furthermore, even if such a case were to occur, centroids from almost-empty clusters can be treated as noisy points in the aggregation step (Alg. 5) on the server. Since they are distant from any true centers, as well as from other received one/many-fit-one centroids, they will be discarded in the grouping process.
>
>
> [1] Structures of Spurious Local Minima in k-means [[paper](https://arxiv.org/pdf/2002.06694.pdf)]

---

> > ### Author Response · Authors · 2024-05-13
> > **Response to Reviewer mSnm**
> >
> > ## * Infinite sample assumption.
> > Infinite-sample assumption is utilized in our theoretical analysis, building upon prior work [1]. However, it is important to note that our algorithm does not depend on this assumption, and we do not impose any data sample size assumption in our experiments. This independence is because structured local solutions persist despite varying data sample size. Therefore, the methods addressing these local solutions within our algorithm remain both effective and necessary. Supplementary experiments on this topic have been incorporated in Appendix C.5 of the revised version.
> >
> > ## * Experiments on varying k are included in Appendix E of the revised manuscript.
> > Thank you for the valuable suggestion. We have included supplementary experiments in the revised manuscript to address this concern. **A detailed discussion is provided in Appendix E of the revised manuscript.**
> >
> > ## * Clarification on notations.
> > Thank you for the constructive suggestion. We will consider it during the revision process to enhance clarity.
> >
> > [1] Structures of Spurious Local Minima in k-means [[paper](https://arxiv.org/pdf/2002.06694.pdf)]

---

> > > ### Comment · Reviewer_mSnm · 2024-05-28
> > > **Response**
> > >
> > > Thank you for providing a detailed response to the raised questions and for making changes to the paper. Most of the questions are answered. However, I still have some:
> > >
> > > - The comparison of Alg3 and Alg4: the authors did not explain why the theoretical version (Alg. 3) cannot be practical. It is supported by theory, therefore I expect it should work in practice as well (otherwise, what is the value of theoretical convergence guarantees if theoretical results cannot be supported by practical evaluation?), but maybe for a more restricted setup. Hence, a more detailed comparison should be added (at least some experiments showing the performance of both versions).
> > >
> > > - Infinite sample assumption: I do not agree with the statement "our algorithm does not depend on this assumption". We discuss the theoretical claims in the paper and they rely on this assumption, and the convergence without it is not provided/impossible to derive. Hence, the convergence theoretically depends on it.

---

> > > > ### Author Response · Authors · 2024-05-30
> > > > **Response to Reviewer mSnm**
> > > >
> > > > We appreciate your valuable feedback and the opportunity for further discussion.
> > > >
> > > > * The comparison of Alg 3 and Alg 4:
> > > >
> > > > The theoretical variant (Alg 3) is designed for a more restricted setup, specifically stochastic ball model assumption. This assumption simplifies the identification of many-fit-one centroids (refer to lines 6-8 in Alg 3). However, this approach may not be applicable beyond these assumptions, particularly as these assumptions typically do not align with practical scenarios. This limitation is the reason why Alg 3 is not deemed practical. Conversely, Alg 4 omits this restrictive step because eliminating many-fit-one in practice is both challenging and unnecessary.
> > > >
> > > > We appreciate your constructive suggestion to provide experiments showcasing the performance of both versions. **We have included a thorough discussion and supplementary experiments in Appendix C.4 of the latest revised version.** These additional results highlight the suboptimal performance of the theoretical variant beyond its assumed conditions, while also emphasizing the effectiveness of our algorithm's empirical version.
> > > >
> > > > * Infinite sample assumption:
> > > >
> > > > Indeed, our theoretical version of the algorithm depends on assumptions like infinite sample and stochastic ball model, while the empirical version has showcased superior performance across diverse scenarios, extending beyond these assumptions. It is important to note that the theoretical analysis serves to provide deeper insights into the algorithm's behavior under specific conditions. **We thank insightful comments and have made modifications to the revised version (refer to Section 4.2).**

---

### Review · Reviewer_GuQL · 2024-04-30

**Summary Of Contributions:**

The paper proposes a novel federated clustering algorithm, FECA (Federated Centroid Aggregation), based on K-Means. First, the algorithm refines the centroids computed with K-Means locally, removing and grouping certain types of centroids. Then, the server is capable of aggregating the centroids in a single training round, i.e., FECA is a one-shot algorithm. Then, the authors also propose an extended algorithm that combines FECA with DeepCluster for unsupervised feature learning with federated learning. The experimental evaluation shows notable improvements with respect to other federated clustering methods in scenarios with varying degree of data heterogeneity.

**Audience:**

Yes

**Claims And Evidence:**

Yes

**Requested Changes:**

See my comments above. Overall, I think that the paper makes some nice contributions, but it would be necessary to make an effort to improve the readability of the paper and to motivate and justify a bit better some of the choices made for the algorithms.

**Strengths And Weaknesses:**

Strengths:
+ Unsupervised learning, in general, and clustering methods in particular have been less explored in the research literature on federated learning. In this sense, the authors tackle an interesting problem, proposing an algorithm capable of learning the centroids in one shot. Overall, the algorithm is reasonable and the extension for unsupervised feature learning with DeepCluster is a nice addition to the paper.
+ The experimental evaluation with synthetic and real datasets is convincing and show a notable improvement with respect to previous state-of-the-art techniques.
+ The authors strived to provide some theoretical analysis on some aspects of the proposed method.


Weaknesses:
+ Although, overall, the algorithms seem reasonable, the motivation and justification of some of them is not very clear and, in my opinion, the narrative of the paper could be improved and there are aspects that require some clarification.


Comments:
+ In Section 3.1 the definition of the cases needs to be more rigorous. For example, what mapping means here? What does it mean that a centroid is mapped to s true centers? The intuition is provided with the figures, but a more formal definition is required.
+ The description of Algorithm 2 is a bit unclear and needs a bit of context. For example, this algorithm usually returns a smaller number of clusters than k, because of the merging and elimination of centroids. This elimination of centroids can have a negative impact on the local solution. However, when reading Section 4.3 (the server aggregation algorithm), the reader can realize that this is not a problem and the global solution returns k clusters. Then, I think it would be convenient to provide more context and clarity throughout the Section 4 to understand better the flow of the algorithms in the clients and the server. This would make the paper more readable.
the one being removed) chosen at random, can help to alleviate the problem.
+ Similarly, Lemma A.1 shows that Algorithm 2 eliminates one-fit-many centroids, but whether the solution after removal of these centroids improves upon the existing solution is unclear. It would be convenient to clarify the message that the authors want to convey there.
+ The justification about why the algorithm works well in scenarios with data heterogeneity is not well supported. In this sense, Algorithm 5 (aggregation of the centroids) is not well justified and discussed.
+ In Algorithm 5, what happens if n < k?
+ In the experiments it looks a bit odd to report the average and the standard deviation in different tables (Table 1 and Table 4).

---

> ### Author Response · Authors · 2024-05-13
> **Response to Reviewer GuQL**
>
> We are grateful to the reviewer for recognizing the novelty of our algorithm. Additionally, we thank the reviewer for constructive comments. We have carefully addressed each point raised and have revised our manuscript accordingly.
>
> ## * Clarify the motivation.
> We appreciate valuable suggestions and have revised Section 1 to provide clarity regarding our motivations.
>
> ## * Clarify the definition in subsection 3.1.
> In response to the reviewer's feedback, we have refined subsection 3.1 to provide a more precise description.
>
> When we mention a 'mapping', we are referring to an association where each centroid from clustering results $\{c_1,\dots,c_k\}$ corresponds to one or more true centers $\{c_1^*,\dots, c_k^*\}$. This association can be classified into two types: one-fit-many and one/many-fit-one. This description is based on [1], which characterizes these associations with error bounds. For better understanding, we provide an intuitive explanation of these associations in subsection 3.1. In the proof of our main theorem, their formal definitions with error bounds [1] are introduced in detail.
>
> ## * Enhanced clarity in algorithmic flow in Section 4.
> We thank the valuable comments and we have revised Section 4 accordingly to improve clarity. Additionally, we have included a detailed flow figure (**Figure 2 in the revised manuscript**) to illustrate the interaction between the algorithms operating on clients and the server.
>
> ## * Ablation study on eliminating one-fit-many centroids is included in Appendix C.3.
> We appreciate the opportunity to clarify the significance of eliminating one-fit-many centroids, and in response, we have included an ablation study in the revised manuscript. In this study, one-fit-many centroids are not eliminated in Algorithm 2 but are sent to the server. The comparative results clearly illustrate a degradation in performance when these centroids are not removed. **For further experimental details, please refer to Appendix C.3 of the revised manuscript.**
>
> ## * Dealing with data heterogeneity.
> The main reason why our algorithm shows promising results in non-IID cases is that the structures of local solutions in k-means remain consistent, despite the heterogeneity of data among clients. As depicted in Figure 1, under non-IID conditions, even if a client converges to the global solution on its own data, this global solution shows similar structures (one-fit-many and many-fit-one) to those observed in local solutions of IID data. Thus, the steps that discard one-fit-many and aggregate many-fit-one centroids remain necessary. This highlights that the methods of resolving structures of local solutions in our algorithm remain both effective and necessary despite the heterogeneity of data.
> **To support this, we have included a detailed discussion on the presence of structured k-means solutions in Appendix C.5.**
>
> ## * If n<k, Algorithm 5 returns n recovered centroids.
> Thank you for raising this point. We have refined Section 4.3 in the revised version for clarification.
>
> On the one hand, it is worth noting that this case is highly unlikely to occur in practical situations when k=k* (k is utilized for clustering on clients, and k* is the number of true clusters). We have never observed this case throughout all our experiments.
>
> On the other hand, let us consider an extreme case where n<k might occur, such as when all clients converge to the same local solution. In this scenario, our algorithm removes one-fit-many centroids from all clients that are associated with the same true clusters. Consequently, no centroids associated with these true clusters will be sent to the server, rendering it impossible for Algorithm 5 to reconstruct corresponding centers without receiving any associated centroids from clients.
>
> Furthermore, it is essential to note that this extreme case is trivial in a federated framework, as all clients would have the same local solutions. Essentially, this case is akin to having only one client encountering a local solution. Therefore, attempting to recover the true centers from one spurious local solution, as in this extreme case, is nearly impossible.
>
> ## * Combined Table 1 and Table 4 in the revised manuscript.
> We appreciate constructive comments and have revised the manuscript accordingly.
>
> [1] Structures of Spurious Local Minima in k-means [[paper](https://arxiv.org/pdf/2002.06694.pdf)]

---

### Decision · Action_Editor_wmpH · 2024-06-12

**Recommendation:** Accept with minor revision

**Comment:**

Overall this paper presents an interesting and effective method for one-shot federated clustering, which would be of interest to the TMLR community. I recommend that the paper be accepted assuming that the revisions discussed above are addressed (i.e., restructuring the paper to discuss the key assumptions earlier on and in more detail, and adding experiments with heterogeneous data to Appendix E).

**Audience:**

All reviewers agreed that the work is of interest to the TMLR community.

**Claims And Evidence:**

All reviewers agreed that the work was generally well-written and the results showed that the proposed method shows significant improvements in the scenarios tested. However, there were some concerns regarding the key assumptions (specifically around the number of instances being infinite, and whether this invalidates the use of FL -- ie.., why would we need a FL algorithm when each client has infinitely many instances?). I would encourage the authors to discuss and justify key assumptions earlier on in the paper---including providing a detailed explanation of the assumptions around the separation conditions referenced from Chen & Xi (2020), which help to motivate the local cluster assignment characterization and the proposed method.

There was also a suggestion to analyze cases with data heterogeneity in the newly included Appendix E, which would help to provide a deeper understanding of the method in heterogeneous settings.

---

> ### Author Response · Authors · 2024-07-08
>
> We would like to extend our sincere gratitude to all the reviewers for their constructive comments, positive feedback, and the time and effort they dedicated to reviewing our work.
>
> For the final version, we have revised the paper to reflect the reviewers' comments. In particular:
> * We have provided a detailed discussion justifying the key assumptions utilized in the theoretical analysis in Section 4 and Appendix A.
> * We have included additional experimental results in Appendix E that explore varying number of $k$ and $k'$ in heterogeneous settings.
>
> We thank again the reviewers for their valuable suggestions. And we remain available for any further questions or clarifications as needed.